# Hybrid algorithms for generating optimal designs for discriminating multiple nonlinear models under various error distributional assumptions

**Ray-Bing Chen[1,2], Ping-Yang Chen[1], Cheng-Lin Hsu[1], Weng Kee Wong[3]***

**1** Department of Statistics, National Cheng Kung University, Tainan, Taiwan, **2** Institute of Data Science, National Cheng Kung University, Tainan, Taiwan, **3** Department of Biostatistics, University of California, Los Angeles, California, United States of America

* wkwong@ucla.edu

## Abstract

Finding a model-based optimal design that can optimally discriminate among a class of plausible models is a difficult task because the design criterion is non-differentiable and requires 2 or more layers of nested optimization. We propose hybrid algorithms based on particle swarm optimization (PSO) to solve such optimization problems, including cases when the optimal design is singular, the mean response of some models are not fully specified and problems that involve 4 layers of nested optimization. Using several classical examples, we show that the proposed PSO-based algorithms are not models or criteria specific, and with a few repeated runs, can produce either an optimal design or a highly efficient design. They are also generally faster than the current algorithms, which are generally slow and work for only specific models or discriminating criteria. As an application, we apply our techniques to find optimal discriminating designs for a dose-response study in toxicology with 5 possible models and compare their performances with traditional and a recently proposed algorithm. In the supplementary material, we provide a R package to generate different types of discriminating designs and evaluate efficiencies of competing designs so that the user can implement an informed design.

## Introduction

Much of the work in optimal design of experiments assumes a known parametric model, apart from the unknown model parameters and the objective is to develop a plan to collect data judiciously for accurate statistical inference. For example, one may wish to design a study to estimate parameters in a nonlinear regression model. In practice, the model is rarely known with certainty and it is likely that there are a few plausible models. Optimal design problems concern identifying the best design, i.e. how to collect data to judiciously select the right model among the plausible models. When there are 2 models and errors are normally distributed and one of the 2 models is fully known, [1] introduced *T*-optimality as a design discrimination

**Data Availability Statement:** They are no real data involved because this work describes how to collect data to identify the right statistical models in the most efficient manner under various criteria

and assumptions. However codes are available to reproduce the results in the paper.

**Funding:** Authors P-YC, R-BC, and WKW received partial support for this study in the form of a grant from the National Institute of General Medical Sciences of the National Institutes of Health under Award Number R01GM107639. R-BC is also partially supported by the Mathematics Division of the National Center for Theoretical Sciences in Taiwan. The funders had no role in study design, data collection and analysis, decision to publish, or preparation of the manuscript.

**Competing interests:** The authors have declared that no competing interests exist.

criterion based on the squared difference between the 2 mean predictions. [2] reviewed optimal discriminating design problems and since then, locally $T$-optimal designs have been applied and studied in various setups, see for example, [3–8] and [9]. When the outcomes are binary [10] or model errors are not normally distributed, [11] proposed $KL$-optimality criterion based on the Kullback-Leibler ($KL$) divergence as the distance measure between the 2 competing models.

Analytical descriptions of optimal discriminating designs rarely exist unless there are simple settings, such as when we want to find an optimal design to discriminate between a constant model and a quadratic model, and both models have homoscedastic errors [1]. When there are multiple models to discriminate, [12] proposed a Fedorov-Wynn type algorithm to find a $T$-optimal design and the convergence of such an algorithm to the optimal discriminating design was established recently under some restrictive conditions [13]. Over time, there were several modifications of the algorithm to find various optimal designs, including [11], who amended it to find $KL$-optimal designs.

Algorithms are a practical way to find optimal discriminating designs. Recently, nature-inspired metaheuristic algorithms have been repeatedly shown to be fast, flexible and efficient for solving hard and high dimensional optimization problems in engineering and computer science. 2 such algorithms are differential evolutionary (DE) algorithm proposed by [14] and particle swarm optimization (PSO) proposed by [15]. [16] was the first to show that PSO outperformed traditional algorithms in statistics for finding a variety of optimal designs. Maximin design problems are much harder problems to solve because the design criterion is non-differentiable and require multiple nested optimization. [17] developed hybridized PSO-based algorithms to solve more complicated optimal design problems such as the standardized maximin optimal criteria, which includes the simpler minimax design problems. Most recently, [18] applied DE to find optimal approximate designs for logistic models with up to 5 factors with all pairwise interaction terms. The number of variables to optimize for such a model is at least 95 if the optimal design is minimally supported; otherwise, there will be many more variables to optimize. For example, if the optimal design has 30 support points, there are 179 variables to optimize.

Our goal is to develop flexible and effective algorithms to solve a broad class of optimal discriminating design problems when there are 2 or more nonlinear models and errors may or may not be normally distributed. Unlike the traditional setup in optimal discriminating design problems, we may not require the null model be fully specified. The work is novel because we apply PSO-based algorithms to solve a broad class of optimal discrimination design problems, including those that require solving 4-level nested optimization problems. Further, we demonstrate that they are more effective than traditional algorithms for finding optimal discriminating designs and also capable of finding optimal discriminating designs that require 4 levels of nested optimization. Commercial statistical software packages do not have programs for finding optimal discriminating designs and there is only one $R$ package for searching specific types of optimal discriminating designs. We develop PSO-based $R$ codes and the reader can freely use them to replicate results in this paper and amend them to solve their optimal discriminating design problems.

Section 2 reviews background, optimal discriminating criteria and search algorithms. In Section 3, we propose 2 algorithms based on PSO to find the optimal discriminating design when there are 2 or more competing nonlinear models with normal or non-normal errors. We also evaluate the performances of the proposed algorithms using several examples. In Section 4, we apply them to construct an optimal design to discriminate among 5 nonlinear models for a toxicology study. Section 5 further demonstrates flexibility and ability of the proposed algorithms to find optimal discriminating designs with singular information matrices and find a

robust discrimination design proposed in [9] where the problem has 4 layers of optimization. In Section 6, we compare efficiencies of the proposed algorithms with a few other algorithms and describe a software package that we have developed for finding a user-selected optimal discriminating design. In addition, we compare the performance of the proposed algorithms with a recent R-package that finds optimal discriminating designs. The last section reinforces the importance and ubiquity of optimal discriminating design problems and contains a summary. The appendix further compares results from both current and the proposed algorithms and the supplementary material contains our R-codes.

## Background

Let $y$ be the univariate response variable and let $f(y \mid x, \theta, \sigma^2)$ be its probability distribution function. The mean response is $\eta(x, \theta)$, where $x$ is an independent variable from a known compact design space $\mathcal{X}$, $\theta$ is an unknown parameter vector and $\sigma^2$ is the variance of $y$, which we may treat as a nuisance parameter. Suppose that there are $K$ models with different underlying probability distributions, $f_1(y \mid x, \theta_1, \sigma_1^2), \ldots, f_K(y \mid x, \theta_K, \sigma_K^2)$, where $\theta_i \in \Theta_i \subseteq R^{m_i}$ for some known positive integers $m_i$, $i = 1, \ldots, K$. Here $\Theta_i$ is the user-selected parameter space for the parameters in the $i^{th}$ model and a compact subspace of the $m_i$-dimensional Euclidean space $R^{m_i}$, $i = 1, \ldots, K$.

Approximate designs were proposed by [19] and they are probability measures defined on $\mathcal{X}$. If an approximate design $\xi$ has support at $s_1, s_2, \ldots, s_n \in \mathcal{X}$ and $p_i$ is its weight at the $i^{th}$ support point $s_i$, we denote it by $\xi = \{s_1, s_2, \ldots, s_n; p_1, p_2, \ldots, p_n\}$ with $\Sigma_i \, p_i = 1$. If the total budget allows for taking a total of $N$ observations for the study, the approximate design $\xi$ takes roughly $Np_i$ observations at the $i^{th}$ support point of $\xi$ subject to each $Np_i$ is an integer and $Np_1 + \ldots + Np_n = N$. When the design criterion is convex (or concave), there are algorithms for finding optimal approximate designs and we can use an equivalence theorem to confirm optimality of a design, including an efficiency lower bound to assess its proximity to the optimum, without knowing the optimum.

### $T$- and $KL$-optimal design criteria

Suppose we have 2 homoscedastic Gaussian models with common variance $\sigma^2$ and different mean functions, $\eta_1(x, \theta_1)$ and $\eta_2(x, \theta_2)$ respectively. Additionally, suppose $\eta_{tr}(x) = \eta_1(x, \theta_{tr})$ is the assumed true model with pre-specified parameter vector $\theta_{tr}$. To discriminate $\eta_{tr}$ from $\eta_2(x, \theta_2)$, [1] proposed the $T$-optimal criterion,

$$T_{2,tr}(\xi) = \min_{\theta_2 \in \Theta_2} \left\{ \int_{\mathcal{X}} \Delta_{2,tr}(x, \theta_2) \, \xi(dx) \right\}, \tag{1}$$

where $\Delta_{2,tr}(x, \theta_2) = [\eta_{tr}(x) - \eta_2(x, \theta_2)]^2$ is the $L_2$-distance between the the mean responses from the 2 models and $\Theta_2$ is a user-specified set. A design $\xi_T^*$ is $T$-optimal if it maximizes (1) over $\Xi$, the set of all designs on $\mathcal{X}$. Because the criterion is concave, optimality of $\xi_T^*$ can be checked using an equivalence theorem based on the directional derivative of the criterion evaluated at the optimum [1]: the design $\xi_T^*$ is $T$-optimal if and only if

$$\psi_T(x, \xi_T^*) = \Delta_{2,tr}(x, \hat{\theta}_2(\xi_T^*)) - T_{2,tr}(\xi_T^*) \leq 0, \tag{2}$$

for all $x \in \mathcal{X}$, with equality at the support points of $\xi_T^*$ and $\hat{\theta}_2(\xi_T^*)$ is the parameter in $\Theta_2$ that minimizes $T_{2,tr}(\xi_T^*)$.

When models do not have homoscedastic or normally distributed errors, [11] proposed the $KL$-optimal criterion to discriminate between them. Suppose $f_1(y \mid x, \theta_1, \sigma_1^2)$ and $f_2(y \mid x, \theta_2, \sigma_2^2)$

are the probability density functions of the 2 competing models and $f_{tr}(y \mid x, \sigma_1^2) = f_1(y \mid x, \theta_{tr}, \sigma_1^2)$ is the true model with a pre-specified $\theta_{tr}$. To measure the difference between the 2 competing models, the criterion uses the Kullback-Leibler (*KL*) divergence given by

$$\mathcal{I}(f_{tr}, f_2, x, \theta_2) = \int f_{tr}(y \mid x, \sigma_1^2) \log \left\{ \frac{f_{tr}(y \mid x, \sigma_1^2)}{f_2(y \mid x, \theta_2, \sigma_2^2)} \right\} \; dy, \; \forall \; x \in \mathcal{X}. \tag{3}$$

The *KL*-optimal criterion of a design $\xi$ is the minimal value of $\mathcal{I}(f_{tr}, f_2, x, \theta_2)$ over $\theta_2 \in \Theta_2$, after the quantity is averaged out with respect to the design $\xi$. We denote this value by

$$I_{2,tr}(\xi) = \min_{\theta_2 \in \Theta_2} \left\{ \int_{\mathcal{X}} \mathcal{I}(f_{tr}, f_2, x, \theta_2) \; \xi(dx) \right\} \tag{4}$$

and the design $\xi_{KL}^*$ that maximizes $I_{2,tr}(\xi)$ among $\Xi$ is the *KL*-optimal for discriminating between $f_{tr}$ and $f_2$. For simplicity, we also reference the assumed known mean response from the true model $f_{tr}$ by "*tr*" and represent $f_2$ by "2" when convenient, as the subscript of $I_{2,tr}$ in (4). Clearly, *T*-optimality is a special case of the *KL*-optimal criterion when errors are homoscedastic and normally distributed. [11] showed that the design $\xi_{KL}^*$ is *KL*-optimal if and only if $\psi_{KL}(x, \xi_{KL}^*)$, the directional derivative of the criterion in the direction of the degenerate design at $x$ evaluated at $\xi_{KL}^*$ satisfies

$$\psi_{KL}(x, \xi_{KL}^*) = \mathcal{I}(f_{tr}, f_2, x, \hat{\theta}_2(\xi_{KL}^*)) - I_{2,tr}(\xi_{KL}^*) \leq 0, \tag{5}$$

for all $x \in \mathcal{X}$ with equality at the support points of $\xi_{KL}^*$. Here $\hat{\theta}_2(\xi)$ is the $\theta_2$ value in (4) that minimizes the *KL* divergence when $\xi = \xi_{KL}^*$.

Most algorithms for finding optimal discriminating designs are based on Fedorov-Wynn type of algorithms and they work well for discriminating between 2 liner models. When there are several nonlinear models, [20] proposed using a weighted sum of the *T* (or *KL*)-optimal criteria values for discriminating between each pair of models in the class along with a New-ton-type algorithm to enhance the search. A potential issue with this approach is that the choice of the weights can be problematic and an improper choice may result in a design having low efficiencies for discriminating between some of the pairs. [21] also proposed max-min optimal discriminating designs for discriminating among 4 logistic models with various pre-dictor functions. By working with 2 models at a time, she modified the algorithm proposed in [20] to maximize the minimum efficiencies across all pairs among all models using a grid of weights. The algorithm took 2,400 seconds to find the maximin *KL*-optimal design.

[6] used nonlinear approximation theory to find *T*-optimal designs and characterized them by considering the maximal absolute difference and not the squared difference between the means of the 2 models. They found that the number of support points could be determined by counting the number of sign changes in the differences between the mean responses over the design space. By taking the absolute value of this difference, they treated the *T*-optimal design problem as a uniform approximation problem and identified those support points in advance. They then calculated the weights for the resulting support points based on the equivalence the-orem. To identify the support points, they used Remes algorithm [22], which is motivated from uniform approximation theory. Based on the sign-changing positions in the difference function, this algorithm alternates the support points iteratively by allocating each one between 2 sign-changing positions. The algorithm stops when all absolute values of the difference at the support points are about the same. The success of their approach depends on the performance of the Remes algorithm, which we will later discuss, including how this and Tommasiś algo-rithm perform relative to the proposed algorithms.

## Maximin *T*- and *KL*-optimal design criteria

In this subsection, we consider the case when there are 3 or more competing models to discriminate. We present discussion for finding maximin KL-optimal designs, with the understanding that when Gaussian models with homoscedastic errors are assumed, the design maximizing (7) below is the max-min *T*-optimal design. [20] and [21] studied the *KL*-optimal discriminating design problems using relative design efficiencies. Without loss of generality, we assume the first model is the true model, $f_{tr} = f_1$, and describe their 2-step approach. First, we identify the *KL*-optimal designs, $\xi^*_{KL,i}$, $i = 2, \ldots, K$, for discriminating between the $i^{th}$ rival model and the true model $f_{tr}$. Given a design $\xi$, the *KL*-efficiency of $\xi$ relative to the *KL*-optimal design $\xi^*_{KL,i}$ is defined by

$$\text{Eff}_i(\xi) = \frac{I_{i,tr}(\xi)}{I_{i,tr}(\xi^*_{KL,i})} \ , i = 2, \ldots, K, \tag{6}$$

where $I_{i,tr}(\xi)$ is given in (4). The optimal discrimination design maximizes the *KL*-efficiencies for all $i$. Therefore, one may find the optimal discriminating design by treating the problem as a multiple objective optimization problem. [20] assumed a pre-specified weight vector, $\alpha = (\alpha_2, \ldots, \alpha_K)$ satisfying $0 \leq \alpha_i \leq 1$ with $\sum_{i=2}^{K} \alpha_i = 1$ is available and proposed finding generalized *KL*-optimal designs that maximize the weighted sum of the *KL*-efficiencies and the vector of weights is $\alpha$. The $i^{th}$ component in $\alpha$ represents the relative importance of identifying the correct model from the $i^{th}$ rival pair of models. If it is problematic to specify $\alpha$, an alternative is to consider the worst possible *KL*-efficiencies [21] and find a design that maximizes the minimal *KL*-efficiency among $\text{Eff}_i(\xi)$, $i = 2, \ldots, K$, i.e. we want a max-min *KL*-optimal design $\xi^*_{mmKL}$ in $\Xi$ that maximizes

$$I_m(\xi) = \min_{2 \leq i \leq K} \text{Eff}_i(\xi). \tag{7}$$

This criterion is concave and we note that the subset $\mathcal{C}(\xi^*_{mmKL})$ comprising the indices of the closest rival model to the true model satisfies:

$$\text{Eff}_i(\xi^*_{mmKL}) < \text{Eff}_j(\xi^*_{mmKL}), \ i \in \mathcal{C}(\xi^*_{mmKL}), j \notin \mathcal{C}(\xi^*_{mmKL}). \tag{8}$$

[21] showed that there is a weight vector $\tilde{\alpha} = (\tilde{\alpha}_2, \ldots, \tilde{\alpha}_K) \in [0, 1]^{K-1}$ that satisfies

$$\sum_{i=2}^{K} \tilde{\alpha}_i = 1 \ \text{ and } \ \tilde{\alpha}_i = 0 \ \text{ if } \ i \notin \mathcal{C}(\xi^*_{mmKL}) \tag{9}$$

such that $\xi^*_{mmKL}$ is the max-min *KL*-optimal design if and only if it is also a generalized *KL*-optimal design with weight vector $\tilde{\alpha}$. The equivalence theorem then states that the design $\xi^*_{mmKL}$ is a generalized *KL*-optimal design if and only if

$$\psi_{mmKL}\left(x, \xi^*_{mmKL}\right) = \sum_{i=2}^{K} \tilde{\alpha}_i \ \frac{\mathcal{I}(f_{tr}, f_i, x, \hat{\theta}_i(\xi^*_{mmKL}))}{I_{i,tr}(\xi^*_{KL,i})} - I_m(\xi^*_{mmKL}) \leq 0, \tag{10}$$

for all $x \in \mathcal{X}$ with equality at all the support points of $\xi^*_{mmKL}$ and

$$\hat{\theta}_i(\xi) = \arg\min_{\theta_i \in \Theta_i} \int_{\mathcal{X}} \mathcal{I}(f_{tr}, f_i, x, \theta_i) \ \xi(dx).$$

To find the maxi-min *KL*-optimal design, [21] proposed a search algorithm based on the equivalence theorems for the max-min *KL*-optimal design and the generalized *KL*-optimal design. The algorithm first searches for a special $\alpha$ vector in (9) so that the generalized *KL*-optimal design corresponds to the sought max-min *KL*-optimal design. [21] implemented

MATHEMATICA codes for the iterative search in a laptop with 2.3 GHz CPU and 4Gb RAM and reported in Section 4 of her paper that the CPU time required to generate the optimal design for discriminating among 4 nonlinear models was about 2,400 seconds, which is expensive. This motivates us to propose an algorithm that avoids the high computational burden for finding the right $\alpha$ vector by searching over a set of user-selected grid points. Our new algorithm uses a metaheuristic algorithm and directly optimizes the max-min $KL$-optimal criterion using a single optimization procedure.

## Hybrid algorithms for finding optimal discriminating designs

Hybridization of 2 or more ways of numerical searches is increasingly common in algorithmic development. The idea is to take advantages of the strengths in the selected algorithms and combine them to solve the optimization problem more effectively than either of the algorithms can. For instance, some algorithms are more effective at determining where the optimum is roughly located (i.e. exploration) and others are more effective at determining the optimum precisely and quickly once it is in its vicinity (i.e. exploitation). The literature is replete with hybrid algorithms and the questions are which is the most appropriate algorithm to hybridize and how to do so.

We now propose hybrid algorithms to find different types of optimal discriminating designs and show that they are generally more effective than current algorithms. The recent successes of using PSO to solve a variety of optimal design problems [16, 17] motivated us to hybridize PSO with another algorithm to find optimal discriminating designs more effectively. After a brief review of PSO, we show how PSO can be hybridized to solve various types of optimal discriminating design problems. These are more challenging design problems than those tackled earlier and as an example, we also apply PSO to solve a complex problem that requires 4 levels of nested optimization.

## Particle swarm optimization

Particle swarm optimization (PSO) is a metaheuristic optimization method proposed by [15]. This nature-inspired algorithm simulates how the birds fly in a coordinated way to look for the optimum, which is where the food is on the ground. Throughout the birds communicate and adjust their velocities and positions iteratively until convergence or the algorithm is terminated by a user-specified stopping rule.

We initiate PSO by generating a flock of $N$ birds (particles) randomly in the given design space. Each particle is a design $\xi$ and we represent it by a vector $(s_1, \ldots, s_n, p_1, \ldots, p_{n-1})^{\top}$, since $p_n = 1 - \sum_{j=1}^{n-1} p_j$. Let $\xi_i^{(t)}$ be the $i^{th}$ particle at the $t^{th}$ iteration. PSO has 2 defining concepts: local best and global best. The design with the maximal design criterion value discovered by the $i^{th}$ particle before the $t^{th}$ iteration is the local best for the $i^{th}$ particle and we denote it by $\xi_{i*}^{(t-1)}$. The global best design is the one found by the whole swarm before the $t^{th}$ iteration and we denote it by $\xi_g^{(t-1)}$. The velocity of the $i^{th}$ particle at the $t^{th}$ iteration is $V_i^{(t)}$ and each particle updates its velocity and position iteratively as follows:

$$V_i^{(t)} = \omega^{(t)} V_i^{(t-1)} + c_1 R_1 \otimes [\xi_{i*}^{(t-1)} - \xi_i^{(t-1)}] + c_2 R_2 \otimes [\xi_g^{(t-1)} - \xi_i^{(t-1)}] \tag{11}$$

and

$$\xi_i^{(t)} = \xi_i^{(t-1)} + V_i^{(t)} \quad \text{for} \quad i = 1, \ldots, N. \tag{12}$$

Here $R_1$ and $R_2$ are 2 independent random vectors whose components are independently drawn from a uniform variate on $[0, 1]$ and the notation $\otimes$ indicates component-wise product.

As with all metaheuristic algorithms, there are tuning parameters. The inertia weight, $\omega^{(t)}$, represents how active the particles are and it is chosen to be a linearly decreasing sequence from 0.95 to 0.2 over the first 80% iterations and fixed at 0.2 for the remaining 20% of the iterations. [15] proposed the parameters $c_1$ and $c_2$ have default values equal to 2 and these choices have been consistently reported to work well in the literature, including [16], who applied PSO to find different types of optimal designs for several biomedical models. [23] provides more details on PSO.

The choice of the initial flock size $N$ is quite arbitrary and likely depends on the size and complexity of the optimization problem. All designs in the flock must have the same number of support points which is usually chosen to be the number of parameters in the mean function, or larger. The typical stopping criterion of PSO is a pre-specified number of the maximum iterations allowed or CPU time or number of function evaluations. Because PSO is quite fast for moderate sized problems and typically converges in a few seconds of CPU time, we can allow a large maximum number of iterations or function evaluations. This also suggests the choice value of $N$ is likely not very important because if the algorithm does not find the optimum, the algorithm can be quickly rerun using another value of $N$. The algorithm PSO can also be terminated when the generated design $\xi_g$ satisfies the equivalence theorem up to a user-specified tolerance or meets the user-specified efficiency lower bound requirement. Algorithm 1 summarizes the basic PSO algorithm.

**Algorithm 1** PSO for finding optimal designs

```
1: Define the design criterion function Φ(ξ, θ), e.g. (1), and Input
   the following: the swarm size N, along with values for the tuning
   parameters (other than the default values)
```

 (1.1.) Generate *initial particles (designs)* $\xi_i^{(0)}$ and velocities $V_i^{(0)}$,
 $i$ = 1, ..., $N$.

 (1.2.) Calculate *design criterion values* $\Phi(\xi_i^{(0)}, \theta)$ for each $i$.

 (1.3.) Initialize *the local and global best designs,* $\xi_{i*}^{(0)} = \xi_i^{(0)}$ and
 $\xi_g^{(0)} = \max_i \xi_{i*}^{(0)}$.

```
2: At the t^th iteration, do
```

 (2.1.) Calculate *particles' velocities* $V_i^{(t)}$ by (11).

 (2.2.) Update *particles* $\xi_i^{(t)}$ by (12).

 (2.3.) Calculate *design criterion values* $\Phi(\xi_i^{(t)}, \theta)$.

 (2.4.) Update *the local best designs* $\xi_{i*}^{(t)} = \max_{s=0,1,...,t} \xi_i^{(s)}$.

 (2.5.) Update *the global best design* $\xi_g^{(t)} = \max_i \xi_{i*}^{(t)}$.

```
3: Output the final global best design ξ_g and Φ(ξ_g, θ).
```

## PSO-QN algorithm for finding an optimal design for discriminating between 2 competing models

We now extend PSO to find *T*- and *KL*-optimal designs when there are 2 competing models. As an illustration, we describe the search for a *T*-optimal design. Given the design space $\mathcal{X}$, the assumed true model $\eta_{tr}(x)$ and the alternative mean function $\eta_2(x, \theta_2)$, our objective is to find a design that satisfies

$$\max_{\xi \in \Xi} T_{2,tr}(\xi) = \max_{\xi \in \Xi} \left\{ \min_{\theta_2 \in \Theta_2} \left\{ \int_{\mathcal{X}} [\eta_{tr}(x) - \eta_2(x, \theta_2)]^2 \, \xi(dx) \right\} \right\}. \tag{13}$$

To find *KL*-optimal designs, we replace the inner objective function in (13) by (4).

There are 2 layers of optimization in this maximin problem with outer and inner optimization problems. To tackle a similar maximin optimization problem, [17] showed their Nested-PSO algorithm was successful in finding different types of maximin optimal designs. The Nested-PSO algorithm utilizes another PSO in Step (2.3) of Algorithm 1 to obtain the fitness value for the outer problem. However, a direct application of the Nested-PSO algorithm to find optimal discrimination designs is computationally demanding and our first proposed algorithm reduces the computational burden by incorporating properties of the optimal discriminating design criteria.

Specifically, we note that the inner objective function in (13) is differentiable with respect to the parameter vector, $\theta_2$ and this implies that we can use derivative-based optimization algorithms, such as Newton's method to obtain the optimization values instead of PSO. We used the limited-memory Broyden–Fletcher–Goldfarb–Shanno (L-BFGS) algorithm which is an extension of Newton's method and widely available, like in a R package `lbfgs` [24] or as a MATLAB function `fminunc`. We also compare its performances and PSO algorithms for solving the inner objective function in (13).

In summary, the proposed algorithm uses PSO to solve the outer problem in (13) with a non-differentiable objective function. Its value found from Step (2.3) in Algorithm 1 is obtained by the L-BFGS algorithm. We call this proposed search strategy the PSO-QN algorithm. Our experience is that the L-BFGS algorithm may fail to work if an improper initial point of $\theta_2$ is chosen. We suggest that when this happens, we randomly choose another initial point and rerun L-BFGS.

We applied PSO-QN algorithm to find an optimal discrimination design when we have 2 rival pharmacokinetic models considered in [11]. The design found by our PSO-QN algorithm is similar to their *KL*-optimal designs; details are in Section A.1 of the appendix.

## PSO-S-QN algorithm for finding a maximin optimal design for discriminating among 3 or more models

This sub-section discuses how we used PSO ideas find the maximin *KL*- (or T-)optimal design, $\xi^*_{mmKL}$. Let the reference model $f_{tr}$ be $f_1$ and let $\xi^*_{mmKL}$ solve the following nested optimization problem:

$$\max_{\xi} \min\left\{ \frac{I_{2,tr}(\xi)}{I_{2,tr}(\xi^*_{KL,2})}, \ldots, \frac{I_{K,tr}(\xi)}{I_{K,tr}(\xi^*_{KL,K})} \right\}$$

where $I_{j,tr}(\xi)$, $j = 2, \ldots, K$, is defined in (4).

To find $\xi^*_{mmKL}$, we apply the PSO-QN algorithm $K-1$ times to identify the *KL*-optimal designs, $\xi^*_{KL,j}$ for each $j = 2, \ldots, K$. These optimal designs are then incorporated into the maxmin *KL*-optimal criterion $I_m(\xi)$ before we solve the 3-layer optimization problem. To solve this optimization problem, we propose modifying Step (2.3) in Algorithm 1 in 2 ways at the $t^{th}$ iteration:

(2.3a). For the $i^{th}$ particle, $\xi_i^{(t)}$, use L-BFGS algorithm to compute $I_{j,tr}(\xi_i^{(t)})$ for $j = 2, \ldots, K$.

(2.3b). Calculate the design criterion value

$$I_m(\xi_i^{(t)}) = \min_{j \in \{2,\ldots,K\}} I_{j,tr}(\xi_i^{(t)})/I_{j,tr}(\xi^*_{KL,j}). \tag{14}$$

We call this modified algorithm PSO-S-QN which also works for finding max-min $T$-optimal designs after replacing the objective function by the $T$-optimality criterion. We note that "S" in PSO-S-QN stands for "screening" because we need to find the minimal one among all the $K-1$ models and the letter "QN" stands for quasi-Newton.

To show the PSO-S-QN-generated design, $\xi_{mmKL}$, is max-min $KL$-optimal, we first identify the model index set $\mathcal{C}$ satisfying (8) where the corresponding efficiency values are minimum among all competing pairs. We then implement a basic PSO algorithm to find the weight vector $\tilde{\alpha}^T = (\tilde{\alpha}_1, \ldots, \tilde{\alpha}_K)$ in (9) by minimizing

$$\int_{\mathcal{X}} \left\{ \sum_{i=2,\ldots,K} \tilde{\alpha}_i \, \frac{\mathcal{I}(f_{tr}, f_i, x, \hat{\theta}_i(\xi_{mmKL}))}{I_{i,tr}(\xi_{KL,i}^*)} - I_m(\xi_{mmKL}) \right\}^2 \xi_{mmKL}(dx)$$

over $\tilde{\alpha}_i \in [0, 1]$ at the support points of $\xi_{mmKL}$ subject to the constraints in (9).

In Sections A.2 and A.3 of the Appendix, we re-visit a couple of max-min optimal design problems for discriminating 3 and 4 models in the literature and demonstrate that the PSO-S-QN algorithms are able to find the same optimal designs or designs that are very close to the reported optimum.

## Application to toxicological experiments

We now apply the PSO-QN and PSO-S-QN algorithms to find an optimal design to discriminate among 5 models in a toxicological study. [25] proposed 5 dose-response models which they found adequate for modelling a continuous endpoint in toxicology. The mean responses from these models are

$$\upsilon_1(x, \theta_1) = a, \qquad\qquad \theta_1 = a > 0, \tag{15}$$

$$\upsilon_2(x, \theta_2) = ae^{-x/b}, \qquad\qquad \theta_2 = (a, b)^\top, a > 0, b > 0, \tag{16}$$

$$\upsilon_3(x, \theta_3) = ae^{-(x/b)^d}, \qquad\qquad \theta_3 = (a, b, d)^\top, a > 0, b > 0, d \geq 1, \tag{17}$$

$$\upsilon_4(x, \theta_4) = a(c - (c-1)e^{-x/b}), \qquad \theta_4 = (a, b, c)^\top, a > 0, b > 0, c \in [0, 1], \tag{18}$$

$$\upsilon_5(x, \theta_5) = a(c - (c-1)e^{-(x/b)^d}), \qquad \theta_5 = (a, b, c, d)^\top, a > 0, b > 0, c \in [0, 1], d \geq 1. \tag{19}$$

All errors are assumed to be independent with mean 0 and homoscedastic, and the design space is user-specified. [25] were interested in how exposure to butyl benzyl phthalate (BBP) in maternal animals during gestation affects the fetal weights. Their study design had eight dose groups with BBP dosages at 0, 270, 350, 450, 580, 750, 970 and 1250 mg/kg body weight/day and 10 female pregnant rats were assigned to each dose. We denote their design by $\xi_{P2000}$ on the dose interval $\mathcal{X} = [0, 1250]$.

[26] used the same study design $\xi_{P2000}$ to illustrate the model selection procedure from this class of models and concluded that model (19) accurately describes the data and the estimated parameters were $\hat{\theta}_5 = (\hat{a}, \hat{b}, \hat{c}, \hat{d}) = (4.282, 835.571, 0.739, 3.515)$. To fix ideas, we assume the largest model (19) is the true model with nominal values given by $\hat{\theta}_5$. Here, we address design issues, i.e. how to judiciously collect observations early on to have a good sense which one of the models is likely be the true model? To this end, we apply the PSO-QN and PSO-S-QN algorithms to search for an optimal design to discriminate among the models, (15) to (19) when errors are normally distributed and when errors are lognormally distributed.

When errors are normally distributed, we use the PSO-QN algorithm to identify all the $T$-optimal designs for discriminating between model (19) and each of the rival models, (15), (16), (17) and (18). The left panel of Table 1 shows the $T$-optimal designs. We then applied PSO-S-QN algorithm to find the max-min $T$-optimal design for discriminating among the models (15)–(19). The max-min $T$-optimal design is $\xi_{mmT}$ = {0.000, 433.345, 1027.333, 1250.000; 0.214, 0.338, 0.249, 0.200} and its $T$-efficiencies relative to each $T$-optimal design are all 77.47%. This implies that $\mathcal{C}(\xi_{mmT}) = \{1, 2, 3, 4\}$. To show that the PSO-S-QN-generated design $\xi_{mmT}$ is max-min $T$-optimal, we calculated the $\tilde{\alpha}$ vector in (9) to be $\tilde{\alpha} = (0.493, 0.000, 0.183, 0.324)$. Fig 1(a) shows the graph of $\psi_{mmKL}$ on the left-hand-side of (10) and confirms the max-min $T$-optimality of the generated design.

When errors are lognormally distributed and the nuisance parameters have a constant coefficient of variation as described in Section A.1 of the Appendix, we follow a similar procedure to find the max-min $KL$-optimal design. Table 1 displays $KL$-optimal designs for pairwise discrimination on the right panel and we observe that they are similar in structure to the $T$-optimal designs. Interestingly, regardless whether the errors are normally distributed or not, the maximum dose of the optimal designs for discriminating between models (19) and (15) and between models (19) and (17) is the largest possible dose allowed, whereas for the other 2 cases, the largest dose in the optimal designs is about the same and equal to about 1064.5. The max-min $KL$-optimal design found by PSO-S-QN algorithm is $\xi_{mmKL}$ = {0.000, 451.530, 1043.591, 1250.000; 0.223, 0.342, 0.248, 0.188} and its $KL$-efficiencies relative to each of the $KL$-optimal designs are all equal to 76.78%. A direct calculation shows the vector $\tilde{\alpha}$ in (9) is (0.504, 0.001, 0.145, 0.350) and the plot in Fig 1(b) confirms its optimality by (10). Our conclusion is that the PSO-S-QN algorithm generated design $\xi_{mmKL}$ is max-min $KL$-optimal.

We now compare our optimal designs with the design $\xi_{P2000}$ with eight doses in [25]. Table 2 shows the $T$- and $KL$-efficiencies for our max-min discrimination designs, $\xi_{mmT}$ and $\xi_{mmKL}$, and $\xi_{P2000}$. The notation $T$−Eff$_j$ is the $T$-efficiency of a design relative to the $T$-optimal design for discriminating between models with mean responses $v_5$ and $v_j$, $j$ = 1, 2, 3, 4; similarly, $KL$−Eff$_j$ is the corresponding $KL$-efficiency. On the left panel of Table 2, the competing models have normally distributed errors and $\xi_{mmT}$ is the best design because its maximized minimal value of $T$-efficiency is 77.47%. If one uses $\xi_{mmKL}$ as the design to discriminate models with normally distributed data, its $T$-efficiency is at least 71.45%. In contrast, the design $\xi_{P2000}$ has less than 60% $T$-efficiency for discriminating any of the other models with model (19). When the pharmacokinetic data is lognormally distributed, the right panel of Table 2 shows the $KL$-efficiency of each design. The performances of the various designs are similar except

**Table 1. The $T$- and $KL$-optimal designs on $\mathcal{X} = [0, 1250]$ when the true model is (19) with nominal values $(a, b, c, d)$ = (4.282, 835.571, 0.739, 3.515).**

| Model Assumption | Normal | Lognormal |
|---|---|---|
| Rival Model | $T$-optimal Design | $KL$-optimal Design |
| (15) | $\left\{\begin{matrix} 0.000 & 1250.000 \\ 0.500 & 0.500 \end{matrix}\right\}$ | $\left\{\begin{matrix} 0.000 & 1250.000 \\ 0.500 & 0.500 \end{matrix}\right\}$ |
| (16) | $\left\{\begin{matrix} 0.000 & 468.156 & 1064.178 \\ 0.249 & 0.498 & 0.253 \end{matrix}\right\}$ | $\left\{\begin{matrix} 0.000 & 487.447 & 1065.370 \\ 0.271 & 0.500 & 0.229 \end{matrix}\right\}$ |
| (17) | $\left\{\begin{matrix} 0.000 & 484.197 & 963.144 & 1250.000 \\ 0.092 & 0.280 & 0.407 & 0.221 \end{matrix}\right\}$ | $\left\{\begin{matrix} 0.000 & 498.900 & 979.719 & 1250.000 \\ 0.093 & 0.290 & 0.407 & 0.210 \end{matrix}\right\}$ |
| (18) | $\left\{\begin{matrix} 0.000 & 468.155 & 1064.177 \\ 0.249 & 0.498 & 0.253 \end{matrix}\right\}$ | $\left\{\begin{matrix} 0.000 & 487.448 & 1065.369 \\ 0.271 & 0.500 & 0.229 \end{matrix}\right\}$ |

(a)

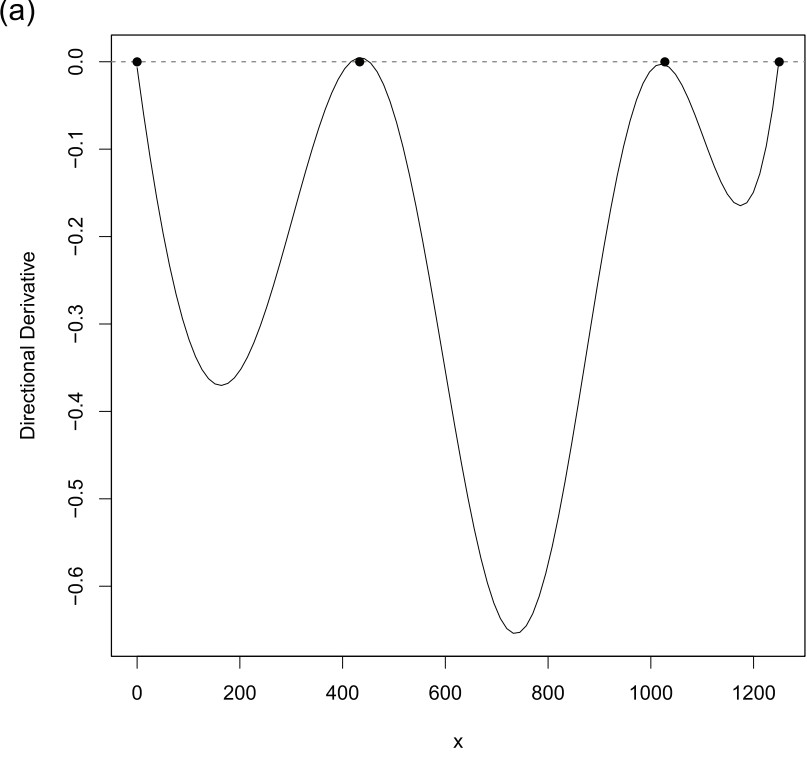

(b)

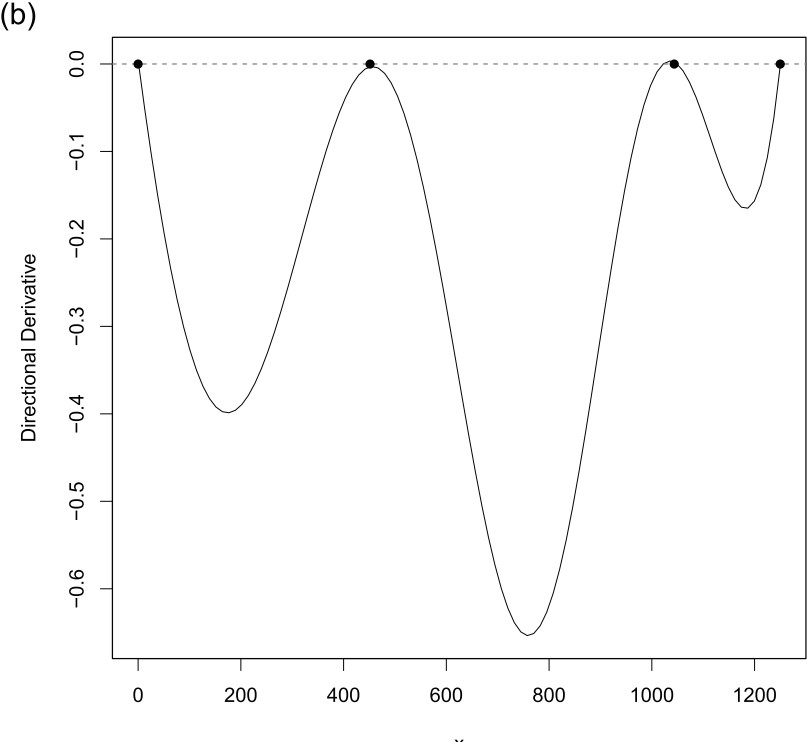

**Fig 1. The plots of $\psi_{mmKL}(x, \xi)$ in (10) for (a) max-min $T$-optimal design $\xi_{mmT}$ and (b) max-min $KL$-optimal design $\xi_{mmKL}$ found by the PSO-S-QN algorithm for discriminating among the 5 toxicology models.** The figures confirm the max-min $T$-optimality and the max-min $KL$-optimality of $\xi_{mmT}$ and $\xi_{mmKL}$, respectively.

**Table 2. $T$- and $KL$-efficiencies of optimal discriminating designs, $\xi_{mmT}$ and $\xi_{mmKL}$, and one selected design, $\xi_{P2000}$.**

| Design | | $T$-efficiency | | | |
|--------|--|------|------|------|------|
| | | $T-\mathrm{Eff}_1$ | $T-\mathrm{Eff}_2$ | $T-\mathrm{Eff}_3$ | $T-\mathrm{Eff}_4$ |
| Max-min $T$-optimal | $\xi_{mmT}$ | 77.47% | 77.47% | 77.47% | 77.47% |
| Max-min $KL$-optimal | $\xi_{mmKL}$ | 76.78% | 80.47% | 71.45% | 80.47% |
| Piersma et al. [25] | $\xi_{P2000}$ | 53.40% | 57.19% | 55.15% | 57.19% |
| Design | | $KL$-efficiency | | | |
| | | $KL-\mathrm{Eff}_1$ | $KL-\mathrm{Eff}_2$ | $KL-\mathrm{Eff}_3$ | $KL-\mathrm{Eff}_4$ |
| Max-min $T$-optimal | $\xi_{mmT}$ | 77.32% | 73.02% | 81.20% | 73.03% |
| Max-min $KL$-optimal | $\xi_{mmKL}$ | 76.78% | 76.78% | 76.78% | 76.78% |
| Piersma et al. [25] | $\xi_{P2000}$ | 52.62% | 53.82% | 54.11% | 53.82% |

for the design $\xi_{P2000}$ which has poor minimal $KL$-efficiency relative to the max-min $T$-optimal design, $\xi_{mmT}$, which has at least 73.02% $KL$-efficiency. These findings suggest that care must be exercised to implement a design to discriminate among a class of models. For this application, it appears that the performances of the various optimal discriminating designs are not much affected whether the errors are normally distributed or not.

## Further examples

We now further demonstrate that the proposed algorithms are flexible and are also able to (i) generate singular optimal discriminating designs, (ii) discriminate models when there are constraints on the model parameters, and (iii) solve discrimination optimal design problem that requires 4 layers of nested optimization over different spaces. For (i), we use an example from [6] and for (ii) we use an example from [1]. [9] proposed robust discrimination designs when there is uncertainty in both the models and their model parameters and we show our algorithms are also able to solve the 4-layer nested optimization problem and produce the same designs as they did analytically.

## Optimal design with singular information matrix

The problem of finding an optimal design to discriminate between a cubic polynomial model and a linear model defined on $[-1, 1]$ was considered in [6]. The mean responses from the 2 models are

$$\eta_1(x) = \gamma_0 + \gamma_1 x + \gamma_2 x^2 + \gamma_3 x^3, \quad \gamma^T = (\gamma_0, \gamma_1, \gamma_2, \gamma_3)$$

and

$$\eta_2(x) = \theta_0 + \theta_1 x,$$

and the vector of nominal values of the parameters in $\eta_1$ is $\gamma = (1, 1, 0, 1)$.

A direct application of the PSO-QN algorithm shows that the $T$-optimal design for this example is $\xi^*_{3pt} = \{-0.500, 0.500, 1.000; 0.167, 0.500, 0.333\}$. This design has 3 unequally supported points and is singular. Its $T$-optimality is confirmed by its directional derivative function (2) plot on the left panel of Fig 2. Clearly, a drawback of this optimal discriminating design $\xi^*_{3pt}$ is that it cannot be used to estimate the 4 parameters in $\eta_1$. The PSO-QN algorithm first searches for the best 4-point design, which is
$\xi^*_{4pt} = \{-1.000, -0.500, 0.500, 1.000; 0.045, 0.211, 0.455, 0.289\}$. Its $T$-optimality is confirmed by the directional derivative plot (2) on the right panel of Fig 2. This design has 4 points

(a)

**3−point T−optimal design for cubic vs. linear models**

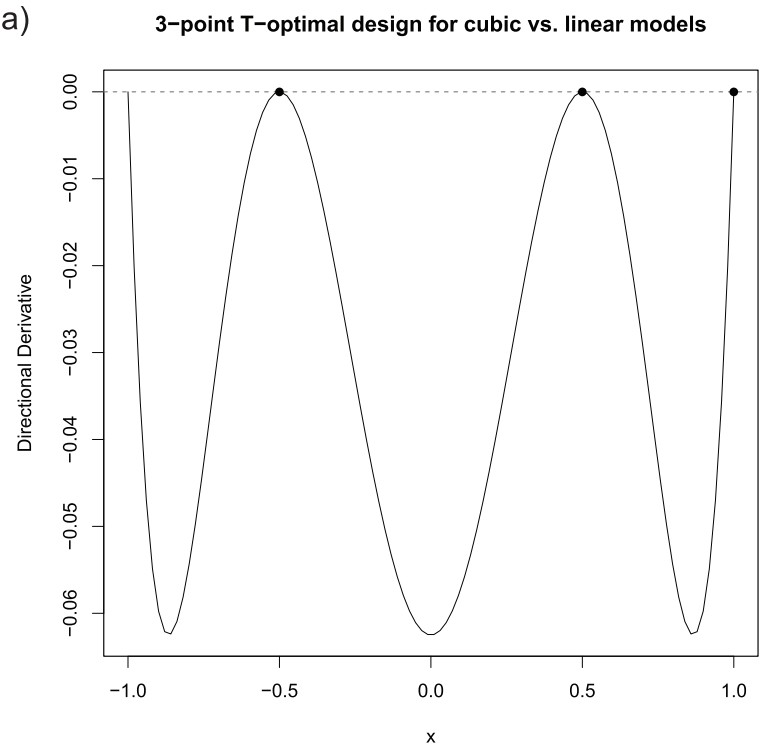

(b)

**4−point T−optimal design for cubic vs. linear models**

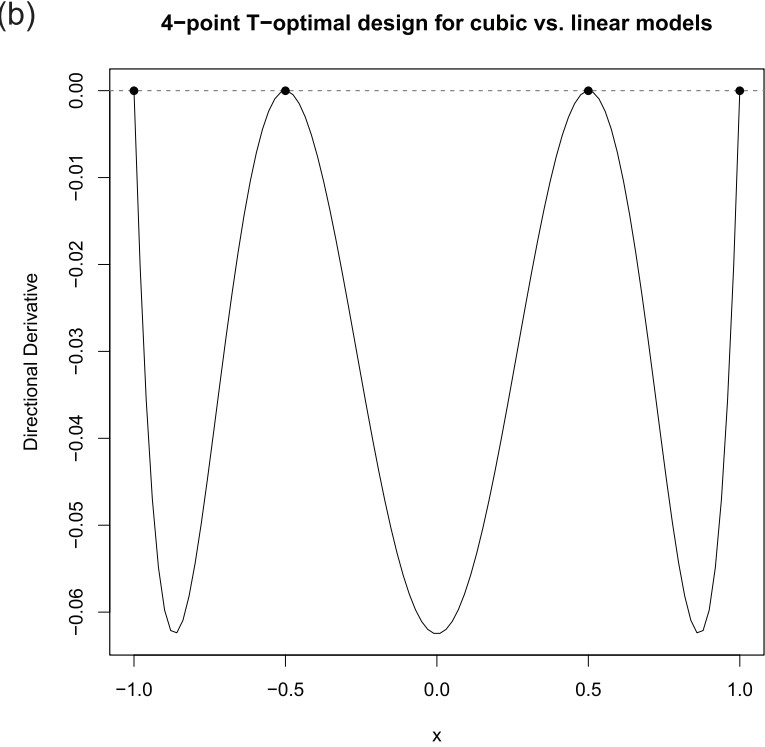

**Fig 2. The plots of $\psi_T(x, \xi)$ in (2) for the cases when the $T$-optimal designs have 3 or 4 points.**

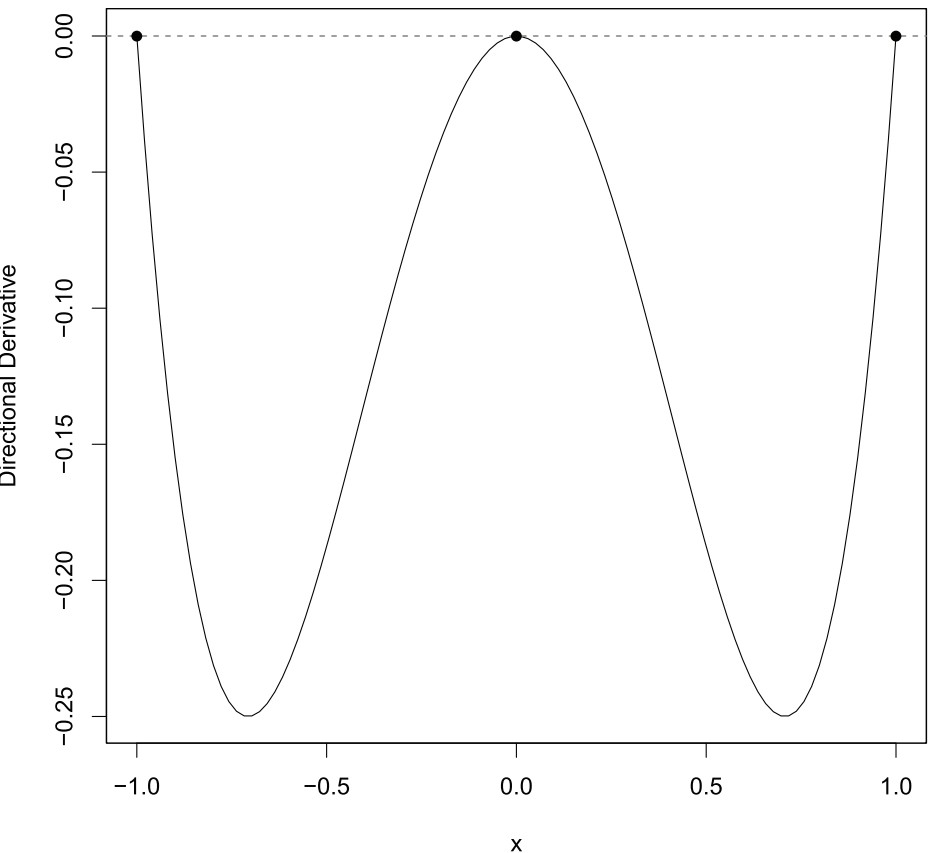

**Fig 3. The plot of $\psi_T(x, \xi)$ in (2) for the case of singular design example in [1].**

and so it can estimate all the parameters in $\eta_1$. Both these designs agree with the designs reported in [6], who also showed that such optimal designs are not unique.

## A larger rival model with a constraint on the model parameters

The design problem to discriminate between 2 models, where the hypothesized true model $\eta_1$ is simpler in structure than the alternative model $\eta_2$ with a constraint on its model parameters was considered in [1]. The 2 models are defined on $[-1, 1]$ and their mean responses are

$$\eta_1(x) = \gamma$$

and

$$\eta_2(x) = \theta_0 + \theta_1 x + \theta_2 x^2,$$

where $\theta_1^2 + \theta_2^2 \geq 1$.

We first transform the constraint in $\eta_2$ to box-type constraint by letting $\theta_1 = r \cos \phi$ and $\theta_2 = r \sin \phi$ where $r \in [1, \infty)$ and $\phi \in [0, 2\pi]$. the PSO-QN algorithm generated the $T$-optimal design $\xi^* = \{-1.000, 0.000, 1.000; 0.25, 0.50, 0.25\}$, which coincides with the $T$-optimal design found in [1]. Fig 3 displays the plot of the directional derivative of the T-optimality criterion evaluated at $\xi^*$ and confirms its optimality.

## Standardized maximin *T*-optimal design

To find *T*- and *KL*-optimal designs, we need to pre-specify the true model with assumed parameter values. However, mis-specified model parameter values can lead to a much less efficient discrimination design. To overcome the mis-specification problem, a robust *T*-optimal criterion was proposed by [9]. Let $\Theta_{tr}$ be a user-selected set containing plausible true values of the model parameters. The strategy is to find a design which is robust to mis-specification of the nominal values of $\Theta_{tr}$. If $\theta_{tr}$ is the vector of model parameters in the true model, the *T*-efficiency of a design $\xi$ is

$$\text{Eff}(\xi, \theta_{tr}) = \frac{T_{2,tr}(\xi, \theta_{tr})}{T_{2,tr}(\xi_T^*(\theta_{tr}), \theta_{tr})}, \text{ where } T_{2,tr}(\xi, \theta_{tr}) = \min_{\theta_2 \in \Theta_2} \Delta_{2,tr}(\xi, \theta_{tr}, \theta_2)$$

and $\xi_T^*(\theta_{tr})$ is the locally *T*-optimal design when the true model has parameter $\theta_{tr}$, i.e. $\xi_T^*(\theta_{tr}) = \arg\max_{\xi \in \Xi} T_{2,tr}(\xi, \theta_{tr})$.. [9] proposed finding a standardized maximin *T*-optimal design, $\xi_{rbstT}^*$, that maximizes the minimal *T*-efficiency, i.e.

$$\xi_{rbstT}^* = \arg\max_{\xi \in \Xi} \left\{ \min_{\theta_{tr} \in \Theta_{tr}} \text{Eff}(\xi, \theta_{tr}) \right\} = \arg\max_{\xi \in \Xi} \left\{ \min_{\theta_{tr} \in \Theta_{tr}} \left\{ \frac{\min_{\theta_2 \in \Theta_2} \Delta_{2,tr}(\xi, \theta_{tr}, \theta_2)}{\max_{\lambda \in \Xi} \min_{\theta_2 \in \Theta_2} \Delta_{2,tr}(\lambda, \theta_{tr}, \theta_2)} \right\} \right\}. \quad (20)$$

To tackle this 4-layer optimization problem, we propose the Nested-PSO-QN algorithm that combines the Nested-PSO in [17] and the PSO-QN algorithm. The outer loop of the Nested-PSO-QN maximizes the minimal *T*-efficiency across the design space and this minimal *T*-efficiency is obtained by searching the interior of the parameter space, $\Theta_{tr}$ in the inner loop. In calculating the *T*-efficiency, we note that the term in the numerator is differentiable and so we used the L-BFGS algorithm to optimize it. The denominator in the *T*-efficiency formula is a locally *T*-optimal design problem, and we had solved it using the PSO-QN algorithm. In the event that the locally *T*-optimal design can be analytically described, the speed of the Nested-PSO-QN algorithm could be accelerated and computation time will be greatly reduced. Below is an example where we used 64 particles and 200 iterations for the outer Nested-PSO-QN loop computation using 64 particles and 50 iterations for the inner Nested-PSO-QN loop computation.

**Example in Dette et al. [9]**. Consider 2 homoscedastic polynomial models defined on $x \in [-1, 1]$ with normally distributed errors and the mean responses are

$$\eta_1(x, \beta) = \beta_0 + \beta_1 x + \beta_2 x^2 + \cdots + \beta_m x^m, \quad (21)$$

and

$$\eta_2(x, \theta_2) = \theta_{20} + \theta_{21} x + \theta_{22} x^2 + \cdots + \theta_{2,m-2} x^{m-2}, \quad m > 2. \quad (22)$$

Here the larger of the 2 nested models is the true model, i.e. $\eta_{tr} = \eta_1$. If $\theta_{tr} = \beta_{m-1}/\beta_m$ and $\theta_{tr} \in \Theta_{tr}$, [9] showed that the problem of finding a standardized maximin *T*-optimal design to discriminate between (21) and (22) is equivalent to that for discriminating between the 2 models with means given by

$$\eta_{tr}(x, \theta_{tr}) = x^{m-1} + \frac{x^m}{\theta_{tr}} \text{ and } \eta_2(x, \theta_2) = \theta_{20} + \theta_{21} x + \theta_{22} x^2 + \cdots + \theta_{2,m-2} x^{m-2}.$$

Suppose the parameter of $\eta_{tr}$ is known to be in the interval $\Theta_{tr} = [-1, 1]$ and we use the Nested-PSO-QN algorithm to find a standardized maximin *T*-optimal design. In this example, the locally *T*-optimal design has a closed-form solution for each $\theta_{tr} \in \Theta_{tr}$ [8] and so we were able to accelerate the Nested-PSO-QN algorithm by incorporating the information into the

**Table 3.** *T*-efficiencies of standardized *T*-optimal designs found by the proposed Nested-PSO-QN algorithm and those reported in [9].

| *m* | Optimal Design | | *T*−Eff |
|---|---|---|---|
| 2 | $\xi_{rbstT}$ | $\begin{Bmatrix} -1.0000 & 0.0022 & 1.0000 \\ 0.3137 & 0.3712 & 0.3151 \end{Bmatrix}$ | 64.00% |
| | $\xi_{DMS2013}$ | $\begin{Bmatrix} -1.0000 & 0.0000 & 1.0000 \\ 0.3125 & 0.3750 & 0.3125 \end{Bmatrix}$ | 64.00% |
| 3 | $\xi_{rbstT}$ | $\begin{Bmatrix} -1.0000 & -0.4161 & 0.4162 & 1.0000 \\ 0.1969 & 0.3031 & 0.3035 & 0.1965 \end{Bmatrix}$ | 61.79% |
| | $\xi_{DMS2013}$ | $\begin{Bmatrix} -1.0000 & -0.4170 & 0.4170 & 1.0000 \\ 0.1973 & 0.3027 & 0.3027 & 0.1973 \end{Bmatrix}$ | 61.79% |

denominator of the *T*-efficiency formula without using the PSO-QN algorithm. We ran the Nested-PSO-QN algorithm for 2 cases when *m* = 2 and *m* = 3 in the above problem. Table 3 displays the standardized maximin designs ($\xi_{rbstT}$), along with the optimal designs $\xi_{DMS2013}$ found by [9], who used a special algorithm to convert the problem to one of finding the root for a Chebyshev's polynomial. The table also displays the various *T*-efficiencies and shows our algorithms were able to produce optimal designs similar to those in [9].

## Implementation, computational efficiency of proposed algorithms and an online tool for finding optimal discriminating designs

We now discuss (i), performances of our algorithms PSO-QN, PSO-S-QN and Nested-PSO-QN relative to other algorithms, (ii) our package for generating a tailor-made optimal discriminating design, and (iii) how to implement our algorithms using C++ codes in a Rcpp package in R [27]. All computations were done on the Linux server with Intel Xeon CPU E5-2620 2.0 GHz and 64GB RAM. In addition, we compare the performance with a R-package which contains 2 functions for the *T*-optimal design and the *KL*-optimal designs.

### Runtime

Table 4 shows the CPU times of one run for all cases investigated in this paper. The computing time for the PSO-QN algorithm depends on the complexity of the model structure. For example, it took only 7 seconds of CPU time to find the optimal design for discriminating between model (19) versus a constant rival model (15). When the rival model is a more complicated model, like model (17), the algorithm took 90 seconds to find the optimal design for discriminating between models (19) and (17).

We expect the PSO-S-QN algorithm requires more time to search for the max-min optimal discriminating designs because we have a 3-layer optimization problem. The total computing time for finding such an optimal discrimination design should becomes noticeably longer when we include time for finding the optimal designs for all the pairwise optimal discriminating design problems. For example, consider the problem, where there are 5 competing models and the PSO-S-QN algorithm was applied to find a max-min *T*-optimal design. We first applied the PSO-QN algorithm to find *T*-optimal designs for discriminating between the assumed true model and each of the rival models (15), (16), (17) and (18). The computing time for searching the *T*-optimal design for each of these 4 2-model discrimination problems was 7.12, 29.02, 90.94 and 73.28 seconds, respectively. We then ran the PSO-S-QN algorithm and it took 399.12 seconds to find the max-min *T*-optimal design. The total computing time

**Table 4. Computing times of one run of the proposed algorithms for all our examples.**

| Algorithm | Number of Layers | Criterion | Section† | Case | CPU time |
|---|---|---|---|---|---|
| PSO-QN | 2 | *T* | | (19) vs. (15) | 7.12 |
| | | | | (19) vs. (16) | 29.02 |
| | | | | (19) vs. (17) | 90.94 |
| | | | | (19) vs. (18) | 73.28 |
| | | | A.2† | (A25) vs. (A26) | 23.11 |
| | | | | (A25) vs. (A27) | 62.38 |
| | | *KL* (Lognormal) | | (19) vs. (15) | 6.84 |
| | | | | (19) vs. (16) | 32.41 |
| | | | | (19) vs. (17) | 90.08 |
| | | | | (19) vs. (18) | 84.52 |
| | | *KL* (Lognormal) | A.1† | (A24) vs. (A23) | 21.12 |
| | | *KL* (Gamma) | | (A24) vs. (A23) | 20.43 |
| | | *KL* (Binomial) | A.3† | (A31) vs. (A28) | 1.39 |
| | | | | (A31) vs. (A29) | 16.75 |
| | | | | (A31) vs. (A30) | 2.41 |
| PSO-S-QN | 3 | max-min *T* | | | 399.12 |
| | | | A.2† | | 165.78 |
| | | max-min *KL* (Lognormal) | | | 423.93 |
| | | max-min *KL* (Binomial) | A.3† | | 85.65 |
| Nested-PSO-QN | 4 | Standardized | | *m* = 2 | 1057.83 |
| | | maximin *T* | | *m* = 3 | 8344.12 |

† Sections A.1, A.2 and A.3 are in the appendix.

for finding the max-min discrimination design is the sum of these computing time which equals 599.48 seconds.

For the standardized maximin *T*-optimal design problems, the Nested-PSO-QN algorithm required 1057.83 and 8344.12 seconds for solving the same problems just discussed when *m* = 2 and *m* = 3, respectively. The computational time for each problem is unsurprisingly long because we were trying to solve 4-layer optimization problems.

## Efficiency of the PSO-QN algorithm

This subsection compares the performance of the PSO-QN algorithm with some well-known algorithms for finding optimal discriminating designs. For *T*-optimal design problems, we consider 2 algorithms, the Fedorov-Wynn algorithm in [1] and the Remes algorithm in [6]. For *KL*-optimal design problems, we consider the Fedorov-Wynn algorithm and also the Nested-PSO algorithm proposed in [17] for solving *T*- and *KL*-optimal design problems.

We used 32 particles and 200 iterations for the PSO-QN and Nested-PSO algorithms and 32 particles and 100 iterations in the inner loop of the Nested-PSO algorithm to minimize the squared difference between the 2 means from the 2 models over the parameter space. For the Fedorov-Wynn type algorithm, we started with a random initial design and pruned the design every 3 iterations during the 200 iterations. For the Remes algorithm, the initial support points were randomly chosen before we ran it for 200 iterations. We implemented them using Rcpp package in R [27] and ran them repeatedly for 50 times by randomly selecting the initial status of the different approaches and computed the efficiencies of the resulting designs relative to the optimal designs.

**Table 5. Performance of various search algorithms for finding *T*-optimal designs.**

| True Model | Rival Model | Search Algorithm | *T*-Efficiency | | | | CPU Time (seconds) |
|---|---|---|---|---|---|---|---|
| | | | Min. | Max. | #(90%+)[†] | #(100%)[‡] | |
| (19) | (15) | PSO-QN | 100.00 | 100.00 | 50 | 50 | 7.59 |
| | | Nested-PSO | 100.00 | 100.00 | 50 | 50 | 200.11 |
| | | Fedorov-Wynn | 97.58 | 99.92 | 50 | 0 | 12.00 |
| | | Remes | 100.00 | 100.00 | 50 | 50 | 1.89 |
| (19) | (16) | PSO-QN | 58.80 | 100.00 | 47 | 47 | 32.00 |
| | | Nested-PSO | 58.81 | 100.00 | 48 | 46 | 213.91 |
| | | Fedorov-Wynn | 91.18 | 99.31 | 50 | 0 | 37.95 |
| | | Remes | 83.00 | 100.00 | 11 | 6 | 11.03 |
| (19) | (17) | PSO-QN | 0.00 | 100.00 | 43 | 43 | 94.38 |
| | | Nested-PSO | 0.00 | 62.38 | 0 | 0 | 232.78 |
| | | Fedorov-Wynn | 89.81 | 95.73 | 49 | 0 | 116.84 |
| | | Remes | 0.00 | 61.18 | 0 | 0 | 15.74 |
| (19) | (18) | PSO-QN | 94.51 | 100.00 | 50 | 49 | 77.85 |
| | | Nested-PSO | 0.04 | 100.00 | 46 | 26 | 216.87 |
| | | Fedorov-Wynn | 0.00 | 100.00 | 49 | 1 | 109.85 |
| | | Remes | 51.73 | 99.82 | 7 | 0 | 12.02 |
| Section A.2 of the appendix | | | | | | | |
| (A25) | (A26) | PSO-QN | 99.67 | 100.00 | 50 | 12 | 22.75 |
| | | Nested-PSO | 0.00 | 4.57 | 0 | 0 | 202.45 |
| | | Fedorov-Wynn | 87.83 | 95.64 | 45 | 0 | 19.35 |
| | | Remes | 0.00 | 99.99 | 48 | 0 | 22.06 |
| (A25) | (A27) | PSO-QN | 100.00 | 100.00 | 50 | 50 | 60.26 |
| | | Nested-PSO | 0.00 | 41.66 | 0 | 0 | 246.63 |
| | | Fedorov-Wynn | 84.90 | 95.95 | 42 | 0 | 44.68 |
| | | Remes | 82.58 | 100.00 | 49 | 5 | 47.15 |

[†]the number of designs with at least 90% *T*-efficiency found over 50 replications.

[‡]the number of *T*-optimal designs found over 50 replications.

Table 5 shows the performances of the 4 algorithms for finding the *T*-optimal designs for the toxicological Experiments and Section A.2 of the Appendix. The results are based on 50 replications and show the range of *T*-efficiency values of the generated designs by different algorithms and the frequencies of their success in finding a design with at least 90% *T*-efficiency. We also report average computing time for each algorithm.

Our overall numerical results show that PSO-QN algorithm outperforms the other 3 algorithms in 5 out of 6 cases in terms of frequency for finding optimal designs. For example, to discriminate between toxicological models (19) and (17), PSO-QN algorithm can find the *T*-optimal design while the rest of the 3 algorithms cannot. For the case of discriminating models (19) and (18), PSO-QN algorithm finds designs with at least 90% *T*-efficiency in all 50 replications and 49 out of them are *T*-optimal. For the same case, Nested-PSO algorithm finds the *T*-optimal design for 26 times; Fedorov-Wynn algorithm and Remes algorithms perform the worst due to low frequency in identifying the optimal design. Only when a simple competing model like model (15) is involved, all algorithms performs similarly. In terms of computational cost, Fedorov-Wynn algorithm and Remes algorithm require shorter computing time than PSO-based algorithms because they start with a single initial design. However, with the same

32 initial designs, PSO-QN algorithm is faster and more efficient than Nested-PSO algorithm. This shows the need for having a specialized algorithm for optimal discrimination design problems.

The Nested-PSO algorithm may not converge when it searches in the inner loop of PSO. One may wonder whether the performance of Nested-PSO depends on the accuracy sought for the optimal solution in the inner loop. Our experience is that the L-BFSG algorithm as the inner loop solver in our PSO-QN algorithm tends to work better. For example, consider the case of discriminating between models (A3) and (A4) in Section A.2 of the appendix. We calculated the inner optimization problem in (1) at the $T$-optimal design, $T_{2,tr}(\xi_{T,2})$, by L-BFGS and PSO algorithms. To have a fair comparison, we terminate both algorithms when the stopping criterion, $|g(t-1) - g(t)|/g(t) < 10^{-6}$, is achieved and $g(t)$ is the value of the objective function at the $t^{th}$ iteration.

We use 4 different swarm sizes in PSO and there are 32, 64, 128 and 256 particles. We ran both algorithms 100 times, each time with a randomly chosen initial value of $\theta_2$, and report the mean value of $T_{2,tr}(\xi_{T,2})$ in Table 6.

Our results suggest that with more particles, PSO is more likely to find the value of $T_{2,tr}(\xi_{T,2})$. This can be seen from Table 6 that shows the standard deviations of the minimal values decreases as the swarm size increases. However, L-BFGS algorithm finds the minimal value, which is smaller than those found by PSO using different swarm sizes. Table 6 also reports the average computing time required for convergence and suggests that L-BFGS algorithm is also faster than PSO. This is a reason that encourages us to use the L-BFGS algorithm to solve the inner optimization problem in (13).

Lastly we compare the performances of the various algorithms for finding $KL$-optimal designs. The Remes algorithm in [6] is not included because we cannot find the details on how to modify the Remes algorithm to find $KL$-optimal designs in their paper. Table 7 shows performances of PSO-QN, Nested-PSO and Fedorov-Wynn algorithms. The results are similar to the previous discussion and suggests that the proposed PSO-QN algorithm is more effective for finding $KL$-optimal discrimination designs since it has the highest frequency for identifying the $KL$-optimal designs in all cases. Fedorov-Wynn algorithm seems adequate for finding highly efficient designs under the $KL$-optimality criterion but seems to have trouble finding the optimal designs. Nested-PSO requires more computing time to find the optimum and its overall performance is not as good as that from PSO-QN.

## Comparison with a R-package

It is instructive to compare performance of the proposed algorithm with other algorithms coded in R for compatibility. After an extensive search, we were only able to find an

**Table 6. Efficiencies of the L-BFGS and PSO algorithms for solving the inner optimization problem (1) in the $T$-optimal design problem in Section A.2 of the appendix.**

| Algorithm | Swarm Size | $T_{2,tr}(\xi_{T,2})$ (Unit: $10^{-3}$) | | | | CPU time (seconds) |
|---|---|---|---|---|---|---|
| | | Min. | Max. | Mean | SD[†] | |
| L-BFGS | – | 1.087 | 1.087 | 1.087 | $\approx 0$ | 0.001 |
| PSO | 32 | 1.106 | 10.500 | 3.614 | 2.413 | 0.056 |
| | 64 | 1.101 | 4.534 | 1.986 | 0.805 | 0.107 |
| | 128 | 1.090 | 3.507 | 1.558 | 0.474 | 0.327 |
| | 256 | 1.091 | 2.241 | 1.280 | 0.234 | 0.601 |

[†]standard deviation.

**Table 7. Performance of various search algorithms for finding *KL*-optimal designs.**

| True Model | Rival Model | Error Assumption | Search Algorithm | KL-Efficiency | | | | CPU Time (seconds) |
|---|---|---|---|---|---|---|---|---|
| | | | | Min. | Max. | #(90%+)[†] | #(100%)[‡] | |
| (19) | (15) | Lognormal | PSO-QN | 100.00 | 100.00 | 50 | 50 | 7.08 |
| | | | Nested-PSO | 100.00 | 100.00 | 50 | 50 | 211.02 |
| | | | Fedorov-Wynn | 99.78 | 99.95 | 50 | 0 | 14.82 |
| (19) | (16) | Lognormal | PSO-QN | 100.00 | 100.00 | 50 | 50 | 32.58 |
| | | | Nested-PSO | 51.30 | 100.00 | 49 | 46 | 229.05 |
| | | | Fedorov-Wynn | 91.29 | 96.39 | 50 | 0 | 43.26 |
| (19) | (17) | Lognormal | PSO-QN | 62.18 | 100.00 | 35 | 35 | 128.83 |
| | | | Nested-PSO | 0.00 | 39.36 | 0 | 0 | 252.82 |
| | | | Fedorov-Wynn | 0.00 | 96.19 | 48 | 0 | 128.69 |
| (19) | (18) | Lognormal | PSO-QN | 99.55 | 100.00 | 50 | 49 | 90.42 |
| | | | Nested-PSO | 14.49 | 100.00 | 45 | 35 | 231.85 |
| | | | Fedorov-Wynn | 0.00 | 96.78 | 48 | 0 | 114.72 |
| Section A.1 of the appendix | | | | | | | | |
| (A24) | (A23) | Lognormal | PSO-QN | 15.22 | 100.00 | 49 | 49 | 20.68 |
| | | | Nested-PSO | 15.26 | 100.00 | 48 | 41 | 209.26 |
| | | | Fedorov-Wynn | 91.64 | 96.11 | 50 | 0 | 18.80 |
| (A24) | (A23) | Gamma | PSO-QN | 15.18 | 100.00 | 48 | 48 | 19.88 |
| | | | Nested-PSO | 69.81 | 100.00 | 46 | 41 | 198.25 |
| | | | Fedorov-Wynn | 92.00 | 97.46 | 50 | 0 | 17.19 |

[†]the number of designs with at least 90% *KL*-efficiency found over 50 replications.

[‡]the number of *KL*-optimal designs found over 50 replications.

appropriate R package called *rodd* for comparison. The R package was published in 2016 and it generates locally and Bayesian optimal discriminating designs [28]. In the *rodd* package, the function, `tpopt`, is for constructing *T*-optimal designs and the function, `KLopt.lnorm`, is for finding *KL*-optimal designs with lognormal errors. These 2 functions were coded based on the algorithms in [29] and [30], respectively. After an initial design is provided, the 2 functions search for an optimal discriminating design using 2 common steps. The first common step is to update the candidate set of the support points by combining the current support points and points that locally maximize $\Phi_T$ or $\Phi_{KL}$. The second common step determines the weights of the candidate support points by maximizing the *T*- or *KL*-criterion directly, and support points with extremely small weights are removed. To speed up the optimization process, a quadratic programming method was proposed and [29, 30] showed that these functions were able to find the optimal discriminating designs after a few iterations.

We report the performances of these 2 functions for searching *T*-optimal designs and *KL*-optimal design for the 5 models, (15)–(19), in the toxicological experiment and errors are lognormally distributed when we consider the *KL*-optimal criterion. We assume model (19) is the true model, as was the case in the earlier comparison section. The tuning parameters in the 2 functions are the same as the default settings in the package. For each function, we ran the algorithm independently 50 times using a specially selected initial design. In the first instance, the initial design was the design equally supported at 10 points generated from Uniform[0, 1250]. For the other 49 instances, the initial design was selected as follows. The number of support points of each of the initial designs was randomly generated from a Poisson distribution with a mean equal to 10. Then we independently sample the required number of support

**Table 8. Performance of `tpopt` function for finding *T*-optimal designs.**

| True Model | Rival Model | *T*-Efficiency | | | | Succ. Trails | CPU Time (seconds) |
|---|---|---|---|---|---|---|---|
| | | Min. | Max. | #(90%+)[†] | #(100%)[‡] | | |
| (19) | (15) | 0.00 | 0.00 | 0 | 0 | 29 | 0.02 |
| (19) | (16) | 0.00 | 100.00 | 16 | 16 | 35 | 0.07 |
| (19) | (17) | 0.00 | 100.00 | 7 | 7 | 31 | 0.11 |
| (19) | (18) | 0.00 | 0.88 | 0 | 0 | 9 | 0.59 |

[†]the number of designs with at least 90% *T*-efficiency found over 50 replications.
[‡]the number of *T*-optimal designs found over 50 replications.

**Table 9. Performance of `KLopt.lnorm` function for finding *KL*-optimal designs.**

| True Model | Rival Model | *KL*-Efficiency | | | | Succ. Trails | CPU Time (seconds) |
|---|---|---|---|---|---|---|---|
| | | Min. | Max. | #(90%+)[†] | #(100%)[‡] | | |
| (19) | (15) | 59.27 | 100.00 | 47 | 44 | 50 | 0.06 |
| (19) | (16) | 8.31 | 100.00 | 44 | 15 | 50 | 0.39 |
| (19) | (17) | 0.00 | 100.00 | 39 | 14 | 50 | 0.81 |
| (19) | (18) | 56.06 | 100.00 | 47 | 1 | 50 | 2.49 |

[†]the number of designs with at least 90% *KL*-efficiency found over 50 replications.
[‡]the number of *KL*-optimal designs found over 50 replications.

points from Uniform[0, 1250], generate a random sample $w_i'$s from Uniform(0,1) and assign weight $w_i / \sum_i w_i$ to the $i^{th}$ support point. The relative *T*- and *KL*-efficiencies are then recorded and compared with other search algorithms. Due to the different initial designs, we also report the frequencies that the function can successfully generate designs without an error message.

Tables 8 and 9 report the comparison results. We observe that the function, `tpopt`, for finding *T*-optimal designs is sensitive to the initial design. In particular, there were only 29 times that the `tpopt` function was able to generate a design without an error message for discriminating between models (15) and (19). In contrast, the other function `KLopt.lnorm` appeared more numerically stable because there was no error message for all the 50 runs and had fast computational time. The design generated by `KLopt.lnorm` frequently had more than 90% design efficiencies, except for the case when we want to discriminate between models (15) and (19), which can be low. In contrast, Table 7 shows the PSO-QN generated designs consistently have higher *KL*-efficiencies in all the 4 cases.

## An open resource in R software for finding optimal discrimination designs

We have devoted much time to develop a software package called **DiscrimOD** for R users to find various types of optimal discrimination designs in this paper. The user can download the file, `DiscrimOD_0.1.1.tar.gz`, from the supplementary material and install the **DiscrimOD** package by the R code, `DiscrimOD_Install.r`. This package allows the user to implement the PSO-QN and the PSO-S-QN algorithms to find the discrimination designs for their own problems. For comparison purposes, we have included both the Fedorov-Wynn and Remes algorithms for finding optimal discrimination designs when there are 2 competing models.

There are previously developed R packages, such as `Rcpp` [27], `RcppDE` [31] and `lbfgs` [24] that have high-end programming techniques and we had incorporated them to make our

software package more flexible and broadly applicable. For instance, the user can input his or her distance measures between 2 models, along with the error distributional assumptions and compute the optimal discriminating design of interest. All the algorithms in the **DiscrimOD** package are built using C++ coding for faster computation. The user only needs basic knowledge of R programming to modify the codes by redefining a function or list object in R. For an advanced R user, one can input the competing models and distance function in C++ codes to accelerate the computation.

We provide R codes for implementing all the examples in this paper and Sections A.1 to A.3 in the Appendix. For example, by running the R codes in `demo_Section_4_tox_T.r`, our package will generate *T*-optimal and max-min *T*-optimal designs for the 5 toxicology models in the section of application to toxicological experiments. Specifically, there are 6 steps:

(#1) define the 5 competing models ([15])–([19]) using the R function object;

(#2) specify the set of nominal values for the parameters in the true model and the parameter space for each rival model;

(#3) define the distance measure function, which is the squared difference, between any 2 models;

(#4) set the values of the tuning parameters for the algorithms;

(#5) use the PSO-QN algorithm to find the *T*-optimal designs for each pair of the models to be discriminated and check their *T*-optimality by the equivalence theorem; and

(#6) use *T*-optimal designs obtained in the previous step and the PSO-S-QN algorithm to find the max-min *T*-optimal design for discriminating among the 5 models, and confirm its max-min *T*-optimality by the equivalence theorem.

Similar to the first case shown in Table 5, we also provide an illustrative set of the R codes that we have implemented in `demo_Section_62_comparison.r`. This file shows how to run PSO-QN, NestedPSO, Fedorov-Wynn and Remes algorithms in R and compare the resulting designs. We also provide the codes to generate the results in Table 6, where we show that the L-BFGS algorithm is more efficient than PSO in solving the inner optimization problem in the *T*-optimal design criterion.

## Summary

Optimal discriminating design problems are common across disciplines. For example, [32] developed an optimal design for model discrimination and parameter estimation for studying population pharmacokinetics in cystic fibrosis patients treated with itraconazole. Their design found optimal sampling times to provide reliable estimates of the population parameters and at the same time, discriminate between 2 competing models. Other examples of optimal discriminating design problems are available in cognitive science [33], psychology [34] and chemical engineering [35], to name a few. These are important optimization problems that are still both theoretically and computationally challenging.

We believe the practical way to solve optimal discriminating design problems in practice is to develop increasingly effective algorithms and make them available to the reader. This paper proposes using nature-inspired metaheuristic algorithms to find these hard to find optimal discriminating designs for the first time and we show that they generally perform as well or outperform current algorithms for finding optimal discriminating designs; the Remes algorithm appears competitive in terms of CPU times, except that in all our examples, it did not find the optimal designs as often as our algorithms. Unlike traditional algorithms, PSO is able to

generate optimal designs neatly without need to periodically collapse clusters of points into distinct points. It is also able to generate singular optimal designs seamlessly. Another advantage of PSO is that is does not require the design space to be discretized, which is helpful for solving high-dimensional optimization problems. We applied our algorithms to a toxicology study and generated a design that optimally discriminates among 5 nonlinear models all with a continuous outcome.

To facilitate practitioners implement the proposed algorithms, we provide as supplementary material, a R package for generating optimal designs in this paper. The user-friendly codes can additionally evaluate efficiencies of other designs and be amended to find tailor-made optimal discriminating designs for user-specified problems.

## Appendix

We re-visit a couple of optimal discriminating problems and demonstrate our algorithms can find the same optimal designs. For all the examples, we set tuning parameters for the proposed algorithms in the following way. For the PSO-QN algorithm to identify the $T$- and $KL$-optimal designs, we employed 32 particles and the stopping criterion was 200 iterations. For the PSO-S-QN algorithm to find max-min $T$- and $KL$-optimal designs, we used 32 particles and 400 iterations. The remaining PSO parameters were the same as what we had set before. In the inner loop of both algorithms, we ran the L-BFGS algorithm for 4 times with randomly chosen initial values to check whether it had converged to the same criterion value. The values of the tuning parameters we used for the L-BFGS algorithm were their default values in [24].

### A.1 2 pharmacokinetic models

[11] constructed $KL$-optimal designs for discriminating between the Michaelis-Menten (MM) model and modified Michaelis-Menten (MMM). The 2 mean functions, respectively, are

$$\eta_1(x, \theta_1) = \frac{V_1 x}{K_1 + x}, \qquad \theta_1 = (V_1, K_1); \qquad (A23)$$

$$\text{and } \eta_2(x, \theta_2) = \frac{V_2 x}{K_2 + x} + F_2 x, \qquad \theta_2 = (V_2, K_2, F_2). \qquad (A24)$$

The variable $x$ is the substrate concentration in an experimental range $\mathcal{X} = [0.1, 5]$. For $j = 1$, 2, the parameters $V_1$ and $V_2$ are the reaction rates at maximal concentration level, and $K_1$ and $K_2$ are the Michaelis-Menten constants that represent the concentrations at which half of the maximum velocity rates are reached for the 2 models. The MMM model generalizes the MM model by adding a linear term with coefficient $F_2$.

In this example, we assumed that the MMM model $\eta_{tr}(x) = \eta_2(x, \theta_2)$ is the true model with nominal values $\theta_2 = (V_2, K_2, F_2) = (1, 1, 1)$. [36] assumed the model errors can have a log-normal or gamma distribution. For such distributions, a common assumption of the nuisance parameters is that the response has a constant coefficient of variation [37]. Let $\sigma_1^2$ and $\sigma_2^2$ be the variances of the random errors in the MM and MMM models, respectively, and assume that $\sigma_1^2/\eta_1 = \sigma_2^2/\eta_2 = 1$. The analytical form of the $KL$-divergence is given in [11].

Table 10 shows the PSO-QN-generated designs $\xi_{KL}$ and their $KL$-optimal criterion values, along with the corresponding designs, $\xi_{LTT2007}$ for the 2 error distributions from [11]. We observe that they are similar. Fig 4 shows the plot for the directional derivative of the criterion evaluated at the generated design for each error distribution and confirms that the PSO-QN

**Table 10. PSO-QN generated *KL*-optimal designs for discriminating between the modified Michaelis-Menten model and Michaelis-Menten model versus corresponding designs $\xi_{LTT2007}$ found by [11] based on a common and constant coefficient of variation for 2 error distributions.**

| Assumption | $\xi_{KL}$ | | | $I_{1,tr}(\xi_{KL})$ | $\xi_{LTT2007}$ | | | $I_{1,tr}(\xi_{LTT2007})$ |
|---|---|---|---|---|---|---|---|---|
| Lognormal | 0.1000 | 1.5690 | 5.0000 | 0.002565090 | 0.1000 | 1.5730 | 5.0000 | 0.002565069 |
| | 0.2940 | 0.5000 | 0.2060 | | 0.2935 | 0.4996 | 0.2069 | |
| Gamma | 0.1000 | 1.5690 | 5.0000 | 0.002564359 | 0.1000 | 1.5730 | 5.0000 | 0.002564341 |
| | 0.2870 | 0.5119 | 0.2011 | | 0.2868 | 0.5116 | 0.2016 | |

generated designs are numerically *KL*-optimal because both graphs have non-positive values with values close to zero at the support points of the generated designs.

## A.2 3 models with normal errors

Suppose we wish to find an optimal design to discriminate among 3 linear models with homoscedastic errors defined on $\mathcal{X} = [-1, 1]$ with mean responses given by

$$\eta_1(x, \theta_1) = \theta_{10} + \theta_{11}e^x + \theta_{12}e^{-x}, \tag{A25}$$

$$\eta_2(x, \theta_2) = \theta_{20} + \theta_{21}x + \theta_{22}x^2, \tag{A26}$$

$$\text{and } \eta_3(x, \theta_3) = \theta_{30} + \theta_{31}\sin\left(\frac{\pi x}{2}\right) + \theta_{32}\cos\left(\frac{\pi x}{2}\right) + \theta_{33}\sin(\pi x). \tag{A27}$$

We assume the true model is $\eta_{tr}(x) = \eta_1(x, \theta_1)$ with nominal values $\theta_1 = (\theta_{10}, \theta_{11}, \theta_{12}) = (4.5, -1.5, -2)$. We first applied PSO-QN algorithm to find *T*-optimal designs for discriminating between the 2 rival pairs of models, (A25) and (A26), and, (A25) and (A27). The *T*-optimal designs are, respectively, given by

$\xi_{T,2} = \{-1.0000, -0.6693, 0.1438, 0.9570; 0.2527, 0.4277, 0.2473, 0.0723\}$ and

$\xi_{T,3} = \{-1.0000, -0.7405, -0.1044, 0.6340, 1.0000; 0.1916, 0.3228, 0.2274, 0.1772, 0.0810\}$,

and their *T*-optimal criterion values are $T_{2,tr}(\xi_{T,2}) = 0.001087$ and $T_{3,tr}(\xi_{T,3}) = 0.005715$. [1] considered discriminating between the first pair only as an example in their work and their *T*-optimal design is the same as ours. Results for the second rival pair are new. Fig 5(a) and 5(b) display plots of the directional derivative of the *T*-optimality criterion evaluated at these designs in the direction of the degenerate design at $x$ and they confirm their *T*-optimality because the graphs satisfy the conditions of the equivalence theorem.

To use the above results to discriminate among 3 models, the first step is to substitute the 2 *T*-optimal designs for discriminating each pair of the rival models and their optimal values into the numerator of

$$\text{Eff}_j(\xi) = \frac{T_{j,tr}(\xi)}{T_{j,tr}(\xi_{T,j})}, \; j = 2, 3,$$

to calculate their *T*-efficiencies required in the PSO-S-QN algorithm. The resulting max-min *T*-optimal design found from the PSO-S-QN algorithm is

$\xi_{T,23} = \{-1.0000, -0.7034, -0.0196, 0.5725, 1.0000; 0.2279, 0.3822, 0.2167, 0.1133, 0.0599\}$

and the max-min *T*-optimal criterion value is $I_m(\xi_{T,23}) = \text{Eff}_2(\xi_{T,23}) = \text{Eff}_3(\xi_{T,23}) = 0.806$.

(a)

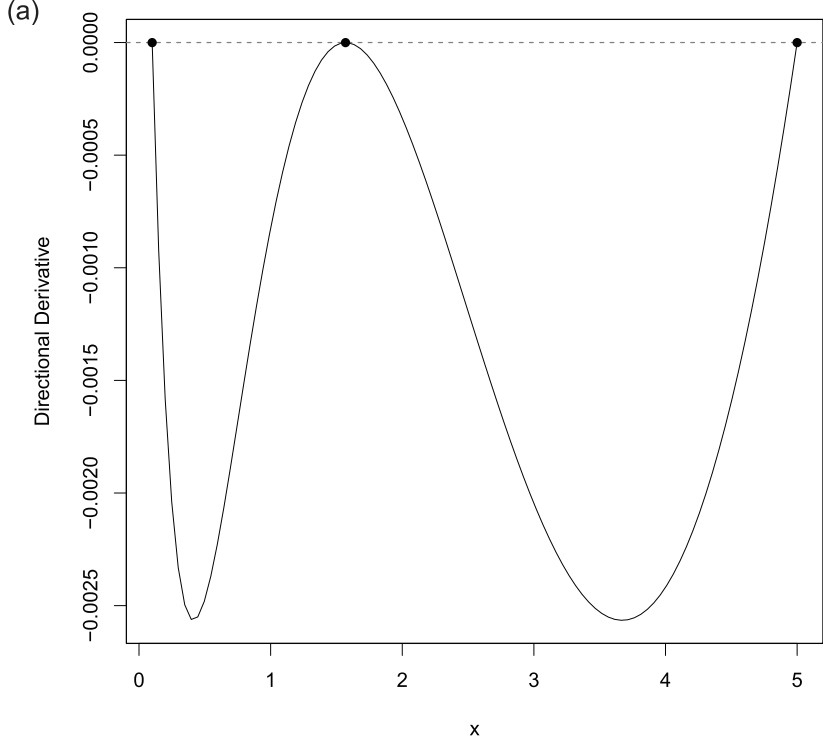

(b)

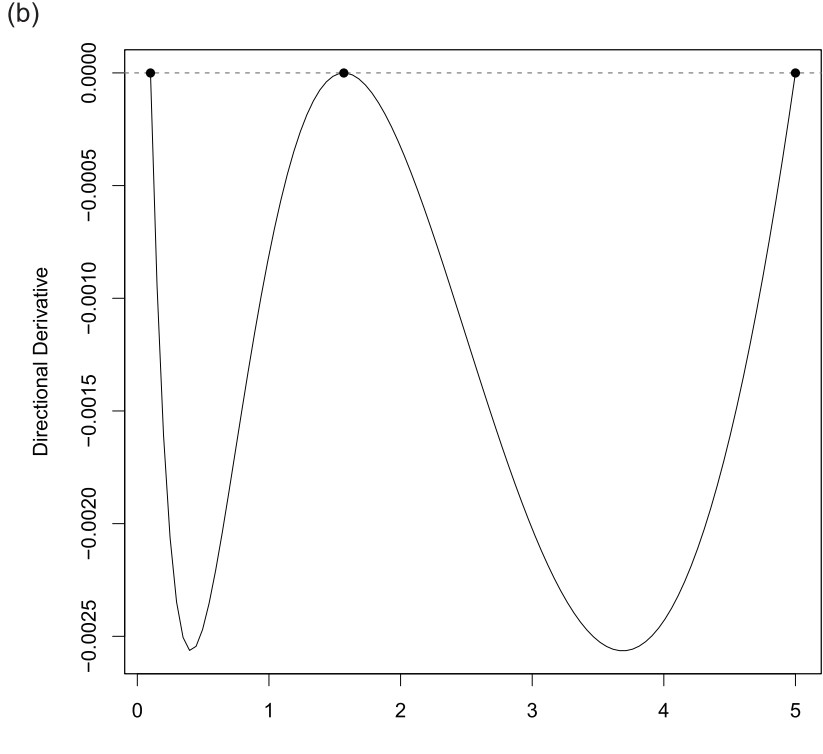

**Fig 4. The plots of the directional derivative of the *KL*-optimality criterion for discriminating between the Michaelis Menten and Modified Michaelis Menten models in the direction of the degenerate design at *x* and evaluated at the PSO-QN-generated designs when errors are (a) lognormal and (b) gamma distributed.** The figures confirm the *KL*-optimality of the 2 designs in Section A.1.

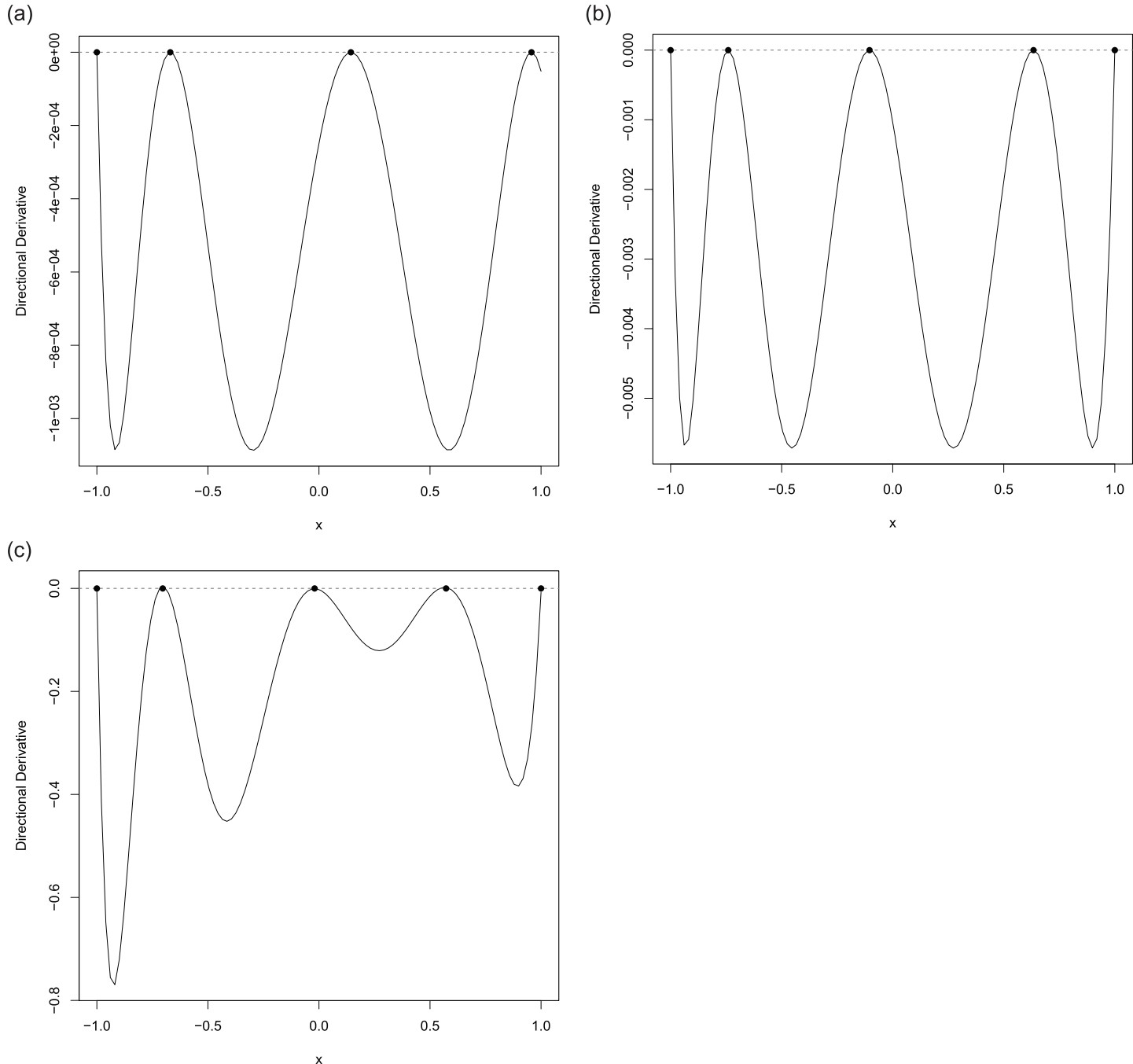

**Fig 5. The plots of the directional derivatives of the optimality criteria evaluated at the generated designs for discriminating between the assumed true model (A25) (a) against model (A26) and (b) against model (A27) and (c) for discriminating among all 3 models.** The figures confirm the $T$-optimality and the max-min $T$-optimality of the 3 designs in Section A.2.

We next show that $\xi_{T,23}$ is max-min $T$-optimal. Our numerical results suggest that the model index set is $\mathcal{C}(\xi_{T,23}) = \{2, 3\}$ and a further application of PSO gives $(\tilde{\alpha}_2, \tilde{\alpha}_3) = (0.688, 0.312)$. Fig 5(c) displays the directional derivative plot of the criterion in the direction of the degenerated design at $x$ and evaluated at the 5-point design $\xi_{T,23}$ and its graph confirms its optimality.

### A.3 Four logistic regression models

[21] considered the design problem for discriminating among 4 logistic models with different regression mean structures:

$$\eta_1(x, \theta_1) = \theta_{10}x, \qquad\qquad\qquad \theta_1 = \theta_{10}, \qquad\qquad (A28)$$

$$\eta_2(x, \theta_2) = \theta_{20} + \theta_{21}x, \qquad\qquad\qquad \theta_2 = (\theta_{20}, \theta_{21}), \qquad\qquad (A29)$$

$$\eta_3(x, \theta_3) = \theta_{30}x + \theta_{31}x^2, \qquad\qquad\qquad \theta_3 = (\theta_{30}, \theta_{31}), \qquad\qquad (A30)$$

$$\text{and } \eta_4(x, \theta_4) = \theta_{40} + \theta_{41}x + \theta_{42}x^2, \qquad\qquad \theta_4 = (\theta_{40}, \theta_{41}, \theta_{42}). \qquad\qquad (A31)$$

It is assumed that the true model is $\eta_{tr}(x) = \eta_4(x, \theta_4)$ with nominal values $\theta_4 = (1, 1, 1)$.

To find the max-min $KL$-optimal design, we first use PSO-QN to find $KL$-optimal designs for discriminating between the true model $\eta_{tr} = \eta_4$ and each of the rival model $\eta_i$, $i = 1, 2, 3$. A direct application of the proposed algorithm produces designs that are similar to those in [21]. Then we apply the PSO-S-QN algorithm and obtain the max-min $KL$-optimal design, $\xi_{KL,123} = \{0.0000, 0.3598, 1.0000; 0.6185, 0.2393, 0.1423\}$. The optimal criterion values is $I_m(\xi_{KL,123}) = 0.619$. We were also able to use the proposed algorithm and reproduce the design, $\xi_{TML2016} = \{0.0000, 0.3615, 1.0000; 0.6184, 0.2391, 0.1425\}$ found by [21]. The optimal value for this design

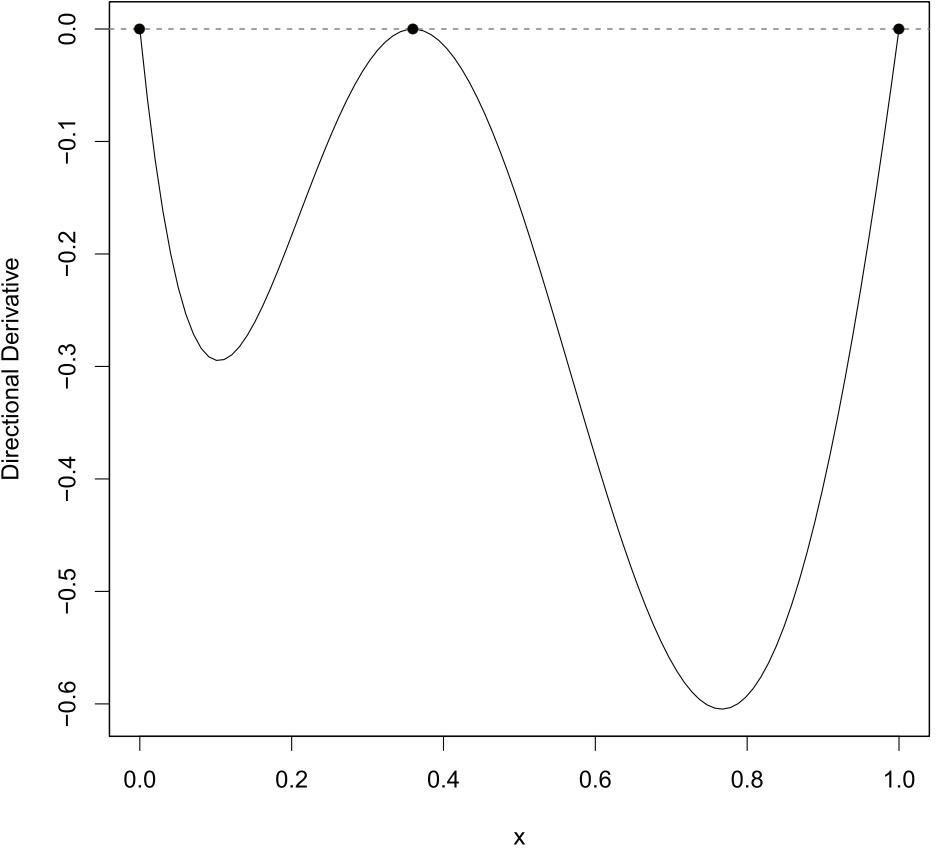

**Fig 6. The directional derivative plot of the PSO-S-QN generated design $\xi_{KL,123}$ for the max-min $KL$-optimal discriminating design problem in Section A.3.** This figure confirms the max-min $KL$-optimality of $\xi_{KL,123}$.

is $I_m(\xi_{TML2016}) = 0.618$ and the $KL$-efficiencies of $\xi_{KL,123}$ relative to the $KL$-optimal designs are

$$I_m(\xi_{KL,123}) = \text{Eff}_{2,tr}(\xi_{KL,123}) = \text{Eff}_{3,tr}(\xi_{KL,123}) = 0.619 < 0.634 = \text{Eff}_{1,tr}(\xi_{KL,123}),$$

which imply that $\mathcal{C}(\xi_{KL,123}) = \{2,3\}$. A further calculation shows the sought vector of $\alpha$ is $\tilde{\alpha} = (\tilde{\alpha}_1, \tilde{\alpha}_2, \tilde{\alpha}_3) = (0, 0.409, 0.591)$ and the directional derivative plot in Fig 6 confirms the max-min $KL$-optimality of $\xi_{KL,123}$.

## Supporting information

**S1 File.**
(RAR)

## Author Contributions

**Conceptualization:** Ray-Bing Chen, Weng Kee Wong.

**Formal analysis:** Ping-Yang Chen.

**Funding acquisition:** Weng Kee Wong.

**Investigation:** Ray-Bing Chen, Ping-Yang Chen, Cheng-Lin Hsu.

**Methodology:** Ray-Bing Chen, Ping-Yang Chen, Weng Kee Wong.

**Project administration:** Ray-Bing Chen.

**Resources:** Ray-Bing Chen.

**Software:** Ping-Yang Chen, Cheng-Lin Hsu.

**Supervision:** Ray-Bing Chen, Weng Kee Wong.

**Visualization:** Ping-Yang Chen.

**Writing – original draft:** Ray-Bing Chen, Ping-Yang Chen, Cheng-Lin Hsu, Weng Kee Wong.

**Writing – review & editing:** Weng Kee Wong.

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
