## [Decision Letter · Decision Letter 0]

27 May 2020

PONE-D-20-10803

Hybrid Algorithms for Generating Optimal  Designs for Discriminating Multiple Nonlinear Models under Various Error Distributional Assumptions

PLOS ONE

Dear Dr. Wong,

Thank you for submitting your manuscript to PLOS ONE. After careful consideration, we feel that it has merit but does not fully meet PLOS ONE’s publication criteria as it currently stands. Therefore, we invite you to submit a revised version of the manuscript that addresses the points raised during the review process.

We look forward to receiving your revised manuscript.

Kind regards,

Ping He, Ph.D., Full Prof., 

Academic Editor

PLOS ONE

Journal Requirements:

2. Please note that PLOS ONE has specific guidelines on software sharing (http://journals.plos.org/plosone/s/materials-and-software-sharing#loc-sharing-software) for manuscripts whose main purpose is the description of a new software or software package. In this case, new software must conform to the Open Source Definition (https://opensource.org/docs/osd) and be deposited in an open software archive. Please see http://journals.plos.org/plosone/s/materials-and-software-sharing#loc-depositing-software for more information on depositing your software.

Additional Editor Comments (if provided):

Please polish your manuscript carefully according to the reviewer's opinion.

Reviewers' comments:

Reviewer's Responses to Questions

**Comments to the Author**

1. Is the manuscript technically sound, and do the data support the conclusions?

Reviewer #1: Yes

Reviewer #2: Partly

Reviewer #3: Yes

Reviewer #4: Yes

Reviewer #5: Yes

Reviewer #6: Yes

Reviewer #7: Yes

2. Has the statistical analysis been performed appropriately and rigorously? 

Reviewer #1: I Don't Know

Reviewer #2: Yes

Reviewer #3: Yes

Reviewer #4: Yes

Reviewer #5: Yes

Reviewer #6: Yes

Reviewer #7: I Don't Know

3. Have the authors made all data underlying the findings in their manuscript fully available?

Reviewer #1: No

Reviewer #2: Yes

Reviewer #3: Yes

Reviewer #4: Yes

Reviewer #5: Yes

Reviewer #6: Yes

Reviewer #7: No

4. Is the manuscript presented in an intelligible fashion and written in standard English?

Reviewer #1: Yes

Reviewer #2: No

Reviewer #3: No

Reviewer #4: No

Reviewer #5: Yes

Reviewer #6: Yes

Reviewer #7: Yes

5. Review Comments to the Author

Reviewer #1: Subject: Review of Manuscript Number: PONE-D-20-10803

Manuscript Title: Hybrid Algorithms for Generating Optimal Designs for Discriminating Multiple Nonlinear Models under Various Error Distributional Assumptions

After carefully reading the article in detail, I came to the conclusion that it was not my field of research, so I cannot review this article accurate completely.

I apologies for this decision.

Reviewer #2: This paper proposes hybrid algorithms based on particle swarm optimization (PSO) to solve such optimization problems when there are multiple models to discriminate, including cases when the optimal design is singular, the mean response of one or more models are not fully specified and problems that involve four layers of nested optimization.The content of the study is interesting.I have several comments about the paper.

1. In this paper, the authors proposed he basic particle swarm optimization algorithm for Generating Optimal Designs for Discriminating Multiple Nonlinear Models.The method used is not novel enough. This algorithm has been out for a long time. The author had better find a new algorithm to solve it.

2.The writing and format of the paper should be improved. It is not clear in several sections. There are some grammatical errors in the article. The author checks them carefully.

3.The introduction is well written and gives an insight of the state of the art. However, to me it fails on giving the reader a realistic scenario in which the proposed routing algorithm could be applied. Introduction is not clear. There are long sentences and they are difficult to understand. Moreover, some ideas are repeated several times. The writing should be also reviewed. There are some typos.

4. The simulation environment parameters are not detailed and the simulation comparison is not deep. It is necessary to add comparative simulation to illustrate the superiority of the proposed method.

Reviewer #3: The authors address the topic of optimal designs for discriminating multiple nonlinear models. The aim of the paper is to design effective algorithms to solve optimal discrimination design problems when there are two or more nonlinear models and errors may or may not be normally distributed. The authors proposed two algorithms based on particle swarm optimization (PSO) to find the optimal discrimination design. Then they did comprehensive experiments to evaluate their proposed algorithms and compared their algorithms with other related algorithms such as Remes algorithm. Based on the analysis and the experiments, the authors asserted that their algorithms are effective and outperform other related algorithms. Overall, the results in the paper are new to the best of my knowledge. However, the manuscript is not very well presented. Thus, I would recommend its publication only if the following issues are addressed.

Main issues:

1) I suggest the authors add “Inputs” in Algorithm 1 to improve readability.

2) The manuscript presents many grammatical errors. Some errors are presented as follows:

In line 73 of page 3, paraameter -> parameter

In line 129 of page 5, propose in Section an algorithm... (add section number)

In line 148 of page 6, propose in Section an algorithm... (add section number)

In line 175 of page 7, cpu time -> CPU time

In line 347 of page 15, For example, we observe in Section ... (add section number)

I recommend the authors to have the manuscript proofread by a colleague.

Reviewer #4: This is an interesting report that addresses optimal discrimination design problems, but I have a number of concerns which I would like to see addressed.

Importantly, please have the manuscript edited by a native English speaker as it contains numerous grammatical and spelling errors

Reviewer #5: Respected Editor

I did considered the manuscript "PONE-D-20-10803"

The manuscript is sound good and contains enough novelty and deserve to be published in Plos one.

Therefore, I recommend to publish it in the current form.

Reviewer #6: A very good effort in optimal solution field. This research will be helpful to select explicit model in variety of solutions where all solutions are seemed correct. The technique used in this article is also helpful for the beginners.

Best of luck

Reviewer #7: Title: Hybrid Algorithms for Generating Optimal Designs for

Discriminating Multiple Nonlinear Models under Various

Error Distributional Assumptions

• Authors investigate different algorithms being used for searching optimal design to discriminate models under various assumptions on error.

• Authors provide a detailed review on T- and KL-optimal Design Criteria and Maximin T- and KL-optimal Design Criteria

• They indicate that run time of the earlier algorithms is not endurable

• They propose hybrid algorithms based on the concept of particle swarm optimization.

Minor Comments

• Page 1 line 3, “apart from the unknown model parameters” should be deleted.

• Page 2 line 3, algorithm should be replaced with algorithms.

• Page 3 line 75, define Өi and Rmi

• Page 9 line 233, homoscedastic should be replaced with homoscedastic error

• Page 11 line 277, “in that they” should be replaced with “and”.

Final Comments

• Overall idea of the paper is good

• Authors claim to have significantly less run time as compared to earlier algorithms

• The R-code or Mathematica code used to develop algorithms should be provided.

Decision

A minor revision of paper is required in the light of following points:

• Authors should correct typos errors

• Paper may be accepted after minor revision and providing R code to verify results.

6. PLOS authors have the option to publish the peer review history of their article (what does this mean?). If published, this will include your full peer review and any attached files.

Reviewer #1: No

Reviewer #2: No

Reviewer #3: No

Reviewer #4: No

Reviewer #5: No

Reviewer #6: No

Reviewer #7: Yes: Tanvir Ahmad

---

## [Decision Letter · Decision Letter 1]

15 Sep 2020

Hybrid Algorithms for Generating Optimal  Designs for Discriminating Multiple Nonlinear Models under Various Error Distributional Assumptions

PONE-D-20-10803R1

Dear Dr. Wong,

We’re pleased to inform you that your manuscript has been judged scientifically suitable for publication and will be formally accepted for publication once it meets all outstanding technical requirements.

Kind regards,

Ping He, Ph.D.

Academic Editor

PLOS ONE

Additional Editor Comments (optional):

The paper can be accepted.

Reviewers' comments:

Reviewer's Responses to Questions

**Comments to the Author**

1. If the authors have adequately addressed your comments raised in a previous round of review and you feel that this manuscript is now acceptable for publication, you may indicate that here to bypass the “Comments to the Author” section, enter your conflict of interest statement in the “Confidential to Editor” section, and submit your "Accept" recommendation.

Reviewer #2: All comments have been addressed

Reviewer #4: All comments have been addressed

Reviewer #6: All comments have been addressed

Reviewer #7: (No Response)

2. Is the manuscript technically sound, and do the data support the conclusions?

Reviewer #2: Yes

Reviewer #4: Yes

Reviewer #6: Yes

Reviewer #7: (No Response)

3. Has the statistical analysis been performed appropriately and rigorously? 

Reviewer #2: Yes

Reviewer #4: Yes

Reviewer #6: (No Response)

Reviewer #7: (No Response)

4. Have the authors made all data underlying the findings in their manuscript fully available?

Reviewer #2: Yes

Reviewer #4: Yes

Reviewer #6: Yes

Reviewer #7: (No Response)

5. Is the manuscript presented in an intelligible fashion and written in standard English?

Reviewer #2: Yes

Reviewer #4: Yes

Reviewer #6: Yes

Reviewer #7: (No Response)

6. Review Comments to the Author

Reviewer #2: The author has revised the paper according to the questions raised by the reviewers. The paper is innovative and suggested to be accepted.

Reviewer #4: Dear authors,

I feel the manuscript is drastically improved.

However, I'm still not sure about the the use of vancouver citation in the body of the text as either a subject or an object of a sentence. I am willing to let this pass.

Reviewer #6: (No Response)

Reviewer #7: (No Response)

7. PLOS authors have the option to publish the peer review history of their article (what does this mean?). If published, this will include your full peer review and any attached files.

Reviewer #2: No

Reviewer #4: No

Reviewer #6: No

Reviewer #7: **Yes: **Tanvir Ahmad

---

## [Editor Report · Acceptance letter]

24 Sep 2020

PONE-D-20-10803R1

Hybrid Algorithms for Generating Optimal Designs for Discriminating Multiple Nonlinear Models under Various Error Distributional Assumptions

Dear Dr. Wong:

I'm pleased to inform you that your manuscript has been deemed suitable for publication in PLOS ONE. Congratulations! Your manuscript is now with our production department.

Kind regards,

on behalf of

Prof. Ping He 

Academic Editor

PLOS ONE